# Neural circuits in the mouse retina support color vision in the upper visual field

Klaudia P. Szatko[1,2,3,7], Maria M. Korympidou[1,3,4,7], Yanli Ran[1,4], Philipp Berens [1,2,5], Deniz Dalkara[6], Timm Schubert[1,4], Thomas Euler [1,2,4] & Katrin Franke [1,2,4 ✉]

Color vision is essential for an animal's survival. It starts in the retina, where signals from different photoreceptor types are locally compared by neural circuits. Mice, like most mammals, are dichromatic with two cone types. They can discriminate colors only in their upper visual field. In the corresponding ventral retina, however, most cones display the same spectral preference, thereby presumably impairing spectral comparisons. In this study, we systematically investigated the retinal circuits underlying mouse color vision by recording light responses from cones, bipolar and ganglion cells. Surprisingly, most color-opponent cells are located in the ventral retina, with rod photoreceptors likely being involved. Here, the complexity of chromatic processing increases from cones towards the retinal output, where non-linear center-surround interactions create specific color-opponent output channels to the brain. This suggests that neural circuits in the mouse retina are tuned to extract color from the upper visual field, aiding robust detection of predators and ensuring the animal's survival.

[1] Institute for Ophthalmic Research, University of Tübingen, Tübingen, Germany. [2] Bernstein Center for Computational Neuroscience, University of Tübingen, Tübingen, Germany. [3] Graduate Training Center of Neuroscience, International Max Planck Research School, University of Tübingen, Tübingen, Germany. [4] Center for Integrative Neuroscience, University of Tübingen, Tübingen, Germany. [5] Institute for Bioinformatics and Medical Informatics, University of Tübingen, Tübingen, Germany. [6] Sorbonne Université, INSERM, CNRS, Institut de la Vision, Paris, France. [7] These authors contributed equally: Klaudia P. Szatko, Maria M. Korympidou. ✉email: katrin.franke@cin.uni-tuebingen.de

Color vision is key to guiding behavior in animals (reviewed in ref. [1]), including navigating in ecological niches (e.g., ref. [2]), communicating with conspecifics (e.g., ref. [3]), foraging as well as detecting predators and prey (e.g., ref. [4,5]). In the retina, signals from different photoreceptor types sensitive to different wavelengths are locally compared by downstream retinal circuits to extract chromatic information present in the visual input (reviewed in ref. [6]). These circuits have been studied in detail in trichromatic primates (reviewed in ref. [6]). Here, signals from short (S; blue), medium (M; green), and long (L; red) wavelength-sensitive cone photoreceptors are processed via two main opponent pathways: red–green (L vs. M) and blue–yellow opponency (S vs. L + M). While the former is mainly based on random and cone-type-unselective wiring of the high-acuity midget system[7–10], blue–yellow opponency relies on precise connectivity in cone-type-selective retinal circuits[11–13].

Compared to primates, the retinal circuits underlying dichromatic vision in other mammals are far from being understood (reviewed in refs. [14,15]). This is also true for the mouse—despite its prominent role as one of today's most frequently used model in visual neuroscience. Mice express S- and M-opsin[16] most sensitive to UV and green light, respectively (Fig. 1a)[17,18]. In addition, M-cones co-express S-opsin, with co-expression increasing towards the ventral retina (Fig. 1b)[19,20]. In contrast, S-cones exclusively expressing S-opsin (true S-cones) make up ~5% of all cones and are homogeneously distributed across the retina[21]. This asymmetric opsin distribution results in a mainly green-sensitive dorsal and a UV-sensitive ventral retina[18]. Nonetheless, behavioral studies have demonstrated that mice can discriminate between light spots of different colors[22], at least in the upper visual field[23]. The retinal circuits underlying this behavior are largely unknown.

Several neuronal circuits for S vs. M color-opponency have been previously proposed in the mouse retina. Some of these circuits involve wiring with S-cone-selective type nine bipolar cells (BCs)[24]. Others do not require cone-type-selective connectivity: For example, alpha retinal ganglion cells (RGCs) located in the vicinity of the horizontal retinal midline exhibit color-opponent responses due to chromatically distinct input to their center and surround[25]. In addition, rod photoreceptors, whose spectral sensitivity closely matches that of M-cones, may also be involved in color-opponency: They provide an antagonistic surround to JAM-B RGCs located in the S-opsin dominated ventral retina by lateral feedback from horizontal cells (HCs)[26]. Such a rod-cone opponent mechanism may support color discrimination in the ventral retina despite the lack of substantial M-opsin expression. While all these studies point at the existence of color-opponent signals downstream from mouse photoreceptors, a comprehensive survey of chromatic processing and the retinal circuits underlying mouse color vision[22,23] is still missing.

Therefore, we systematically investigated the basis for color vision in the mouse retina across three consecutive processing stages. We recorded the output signals of cones, BCs and RGCs to chromatic visual stimulation in the ex vivo, whole-mounted retina using two-photon calcium and glutamate imaging. Surprisingly, we found that across all processing layers, color-opponency is largely confined to the S-opsin dominated ventral retina. Here, color-opponent responses are already present at the level of the cone output, mediated by input from HCs and likely involving rod photoreceptors. We further show how BCs forward the chromatic signals from photoreceptors to the inner retina, where different RGC types integrate information from their center and surround in a type-specific way, thereby increasing the diversity of chromatic signals available to the brain.

## Results

**Recording chromatic cone responses in the mouse retina.** To characterize chromatic signaling in cones, we recorded synaptic glutamate release from their axon terminals. To this end, we expressed the glutamate biosensor iGluSnFR[27] ubiquitously in the retina using a viral approach (Fig. 1c)[28]. In the outer plexiform layer (OPL), where the cone axon terminals are located, this approach resulted in iGluSnFR being predominantly expressed in HC processes[29], which are postsynaptic to the photoreceptors. To identify functional release units, we defined regions of interest (ROIs) using a correlation-based approach (Fig. 1d,i; Methods). These functionally defined ROIs formed a regular mosaic within individual scan fields (Supplementary Fig. 1a–h), reminiscent of the mosaic of EM-reconstructed cone axon terminals[30,31]. In addition, the ROIs colocalized with anatomical cone axon terminals visualized using Sulforhodamine-101 (SR101; Supplementary Fig. 1i)[29]. Together, this suggests that our ROIs correspond to individual cone axon terminals and that densely packed rod photoreceptors—the only other source of glutamate release in the outer retina—appear not to contribute detectably to the glutamate signals recorded in the OPL (Supplementary Discussion). For simplicity, we will in the following refer to ROIs in OPL scan fields as cones.

In total, we recorded light-evoked glutamate responses from 2945 cones ($n = 52$ scan fields, $n = 9$ mice) located in dorsal and ventral retina using full-field (700 μm in diameter) as well as center (150 μm in diameter) and surround (annulus; full-field—center) green and UV light flashes (Fig. 1e, g; Methods). For each cone that passed our quality criterion (Methods), we quantified the chromatic preference of full-field, center, and surround responses by estimating the spectral contrast (SC; for UV- and green-sensitivity SC < 0 and SC > 0, respectively). To sufficiently stimulate all cones within one scan field, the size of the center stimulus was slightly larger than the size of the scan field.

**Ventral cone photoreceptors display color-opponent responses.** We found that the chromatic preference of cone full-field and center responses largely matched the opsin expression across the mouse retina. Generally, as vertebrate cones are Off cells and hyperpolarize upon an increase in light, cone center and full-field responses were characterized by a decrease in glutamate release (Fig. 1e, g). In agreement with the predominance of M-opsin in the dorsal retina, the majority of dorsal cones displayed a strong response to green full-field and center flashes (Figs. 1e–h, 2a, b; $SC_{center} = 0.38 \pm 0.44$, $SC_{full\text{-}field} = 0.37 \pm 0.45$). Due to the long sensitivity tail of M-opsin to shorter wavelengths (cf. Fig. 1a), most dorsal cones showed a small additional response to UV. In addition, consistent with the homogeneous distribution of S-cones[21], a small number of dorsal cones responded strongly to UV light (see e.g., cone (2) in Fig. 1e–h). Ventral cones exhibited UV-dominant responses to full-field and center flashes (Figs. 1j–m, 2a, b; $SC_{center} = -0.7 \pm 0.43$, $SC_{full\text{-}field} = -1.12 \pm 0.43$), as expected from the co-expression of S-opsin in ventral M-cones (e.g., ref. [19]).

In contrast to full-field and center responses, the chromatic preference of cone surround responses did not strictly follow the opsin distribution across the retina. We focused on antagonistic responses where center and surround stimuli result in decrease and increase in glutamate release, respectively. We found that many dorsal cones showed a stronger increase in glutamate to green than to UV surround stimulation (Figs. 1g, h, 2c; $SC_{surround} = 0.39 \pm 1.02$; $n = 216/671$), matching the spectral preference of center and full-field responses. Most ventral cones showed an increase in glutamate solely to green surround stimuli (Fig. 1l, m, 2c; $SC_{surround} = 1.2 \pm 0.42$; $n = 841/1337$), contrasting

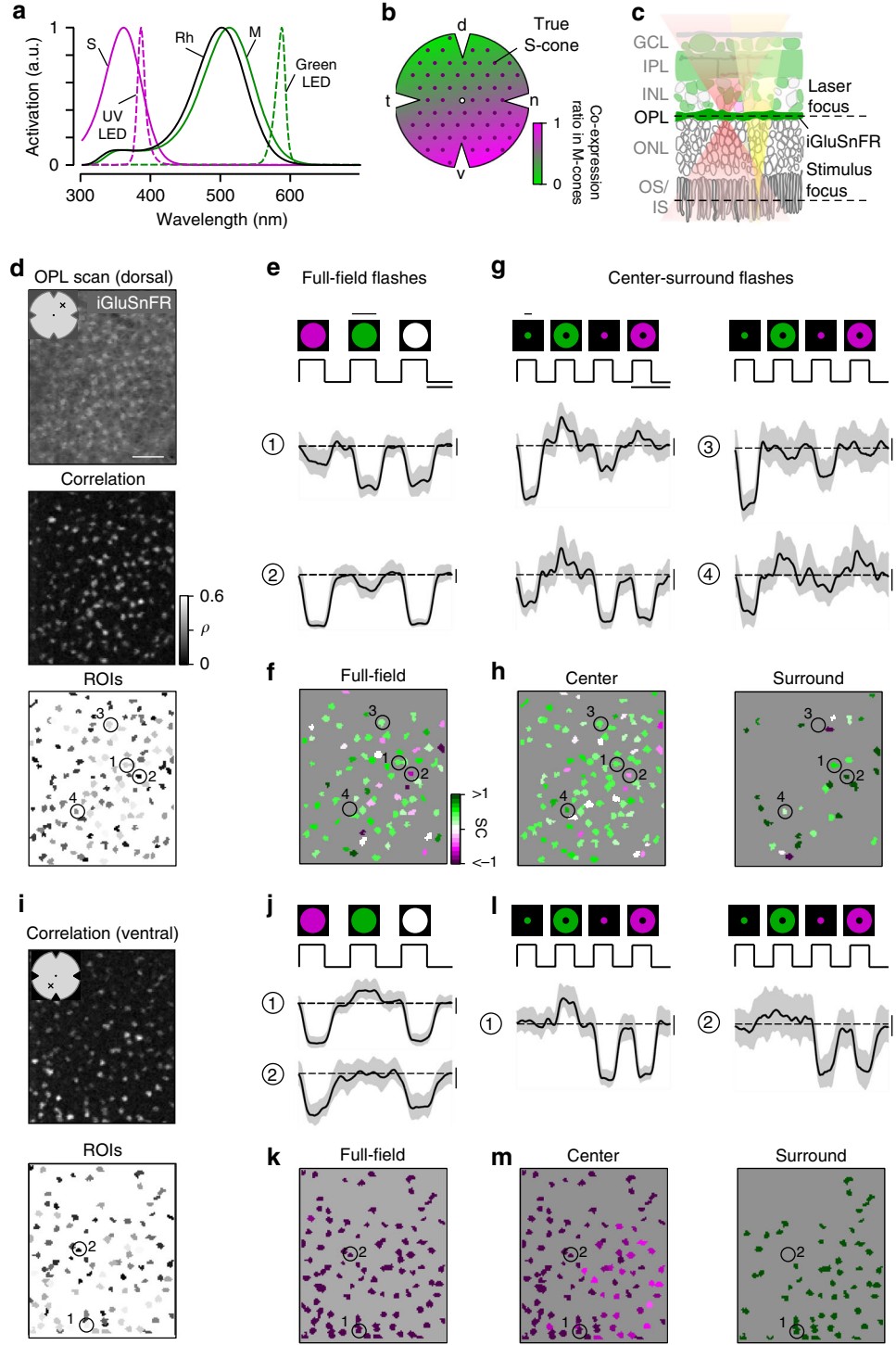

their UV preference for center and full-field responses. This resulted in color-opponent center-surround receptive fields (RFs) and color-opponent full-field responses (Figs. 1j, k, 2a; $SC_{\text{full-field}} < -1$; $n = 937/1329$). A small number of these color-opponent cones in addition showed a slight increase in glutamate upon a green center flash (see left glutamate trace in Fig. 2b), suggesting that center stimulation may be sufficient to drive color-opponent responses in some cones. Surprisingly, UV-sensitive cones in ventral and dorsal scan fields consistently showed the same response polarity for both UV center and surround stimulation

(see e.g., cones (1) and (2) in Fig. 1l), which might be due to increased scattering of UV light (Supplementary Discussion).

Next, we investigated the origin of the green surround responses in the ventral retina. As S-opsin expression strongly increases towards the ventral retina, the main source of green sensitivity should be rod photoreceptors (cf. Fig. 1a). Recently, it has been proposed that rod signals are relayed to cones via HCs[26]. To test this hypothesis, we recorded cone responses to chromatic center-surround stimuli while blocking light-evoked HC feedback using NBQX, an antagonist of AMPA/kainate-type glutamate

**Fig. 1 Imaging chromatic signals from cone axon terminals in the mouse retina. a** Sensitivity spectra of mouse S- (magenta) and M-opsin (green) and rhodopsin (black; Rh), with emission spectra of UV (magenta, dotted) and green LED (green, dotted). **b** Schematic distribution of cone photoreceptors across the mouse retina. Dots and shading represent distribution of true S-cones and co-expression ratio of S- and M-opsin in M-cones, respectively. d dorsal, n nasal, v ventral, t temporal. **c** Schematic experimental setup for cone recordings. OS/IS outer/inner segment, ONL outer nuclear layer, OPL outer plexiform layer, INL inner nuclear layer, IPL inner plexiform layer, GCL ganglion cell layer. Red and yellow shading illustrate laser and stimulus beam, respectively. **d** Example scan field (93 × 110 μm, 3.9 Hz) located in dorsal retina, showing iGluSnFR expression in the OPL (top, scale bar: 20 μm), correlation image (middle) and ROI mask (bottom). For display, the light artifact on the left side of scan fields was cut, resulting in 108 × 128 pixels (instead of 128 × 128). **e** Cone responses of exemplary ROIs from (**d**, bottom) to full-field UV, green and white flashes (700 μm in diameter, scale bar: 1 s). As vertebrate photoreceptors are Off cells, light responses correspond to a decrease in glutamate release. Traces show mean glutamate release with s.d. shading (n > 10 trials, error bars: 1 s.d.). Dotted line indicates baseline. Traces scaled according to s.d. of baseline. **f** Cells from (**d**, bottom) color-coded according to their SC in response to full-field flashes. **g** Glutamate traces of cells from (**d**, bottom) in response to UV and green center (150 μm in diameter) and surround flashes (scale bar: 2 s). **h** Cells from (**d**, bottom) color-coded based on center (left) and surround SC (right). **i** Correlation image (top) and ROI mask (bottom) for an exemplary scan field located in the ventral retina. **j–m** Like (**e–h**), but for cells shown in (**i**, bottom). Scan fields/traces shown in this figure correspond to representative examples. In total, we recorded n = 52 scan fields in n = 9 mice. For quantification, see Fig. 2.

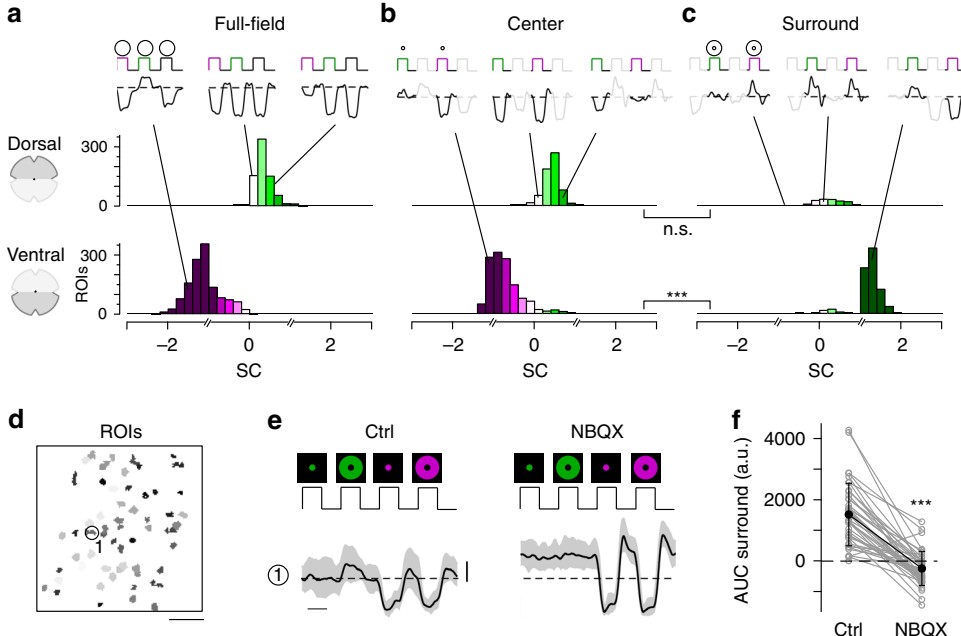

**Fig. 2 Differential chromatic processing in ventral and dorsal cones. a–c** Histograms of spectral contrast (SC) of dorsal (top) and ventral (bottom) cones in response to full-field (**a**, n = 791/1329 dorsal/ventral ROIs), center (**b**, n = 671/1337 dorsal/ventral ROIs), and surround (**c**, n = 216/841 dorsal/ventral ROIs) flashes. Mean glutamate traces of single cones above histograms illustrate the diversity of cone responses to chromatic stimuli. Interestingly, only a fraction of cones showed antagonistic surround responses characterized by an increase in glutamate release (dorsal retina: n = 689/922 cones; ventral retina: n = 1344/2181 cones, n = 52 scan fields, n = 9 mice; Supplementary Discussion). Breaks in the x-axis indicate where different equations for estimating SC were used (Methods). n.s. not significant (p = 0.053); ***p = 2.2 × 10$^{-16}$; linear mixed-effects model for partially paired data (for details, see Methods and Supplementary Methods). **d** ROI mask of scan field located in ventral retina. Scale bar: 20 μm. **e** Glutamate traces of exemplary cone from **d** in response to UV and green center and surround flashes under control and drug condition (50 μM NBQX). Traces show mean glutamate release with s. d. shading (n > 10 trials; error bar: 1 s.d.). Traces scaled according to s.d. of baseline. Scale bar: 1 s. **f** Effect of bath-applied NBQX on area under the curve (AUC) of green-sensitive surround responses of ventral cones (n = 40 ROIs, n = 3 scan fields, n = 2 mice). Error bars show ±s.d. ***p < 0.001; Wilcoxon signed-rank test for paired data.

receptors (Fig. 2d, e; see e.g., ref. [29]). This caused a significant decrease in green-sensitive surround responses in ventral cones (Fig. 2f), confirming that HCs contribute to generating color-opponent responses in cones. In addition, the disrupted HC feedback disinhibited the cones and thereby increased the baseline level of glutamate release, which is in-line with previous studies (e.g., ref. [32]).

In summary, we found that the chromatic preference of a cone's center and full-field response mirrored the overall opsin distribution at the recording site, with largely UV- and green-sensitive responses in the ventral and dorsal retina, respectively. However, while in dorsal cones the chromatic preference of center and surround was very similar, ventral cones systematically

exhibited a strong green-shift in the chromatic preference of their antagonistic surround, likely involving HC input driven by rods. This results in color-opponent responses in most ventral cones, demonstrating that color-opponency is already present at the first synapse of the mouse visual system.

**Bipolar cells relay chromatic information to the inner retina.** Next, we investigated how the chromatic information present in the cone output is relayed to the inner retina by the BC population. In the mouse retina, the signals from photoreceptors are distributed among 14 BC types[28,30,33], with their axonal arbors stratifying at different levels of the inner plexiform layer (IPL)[30,34].

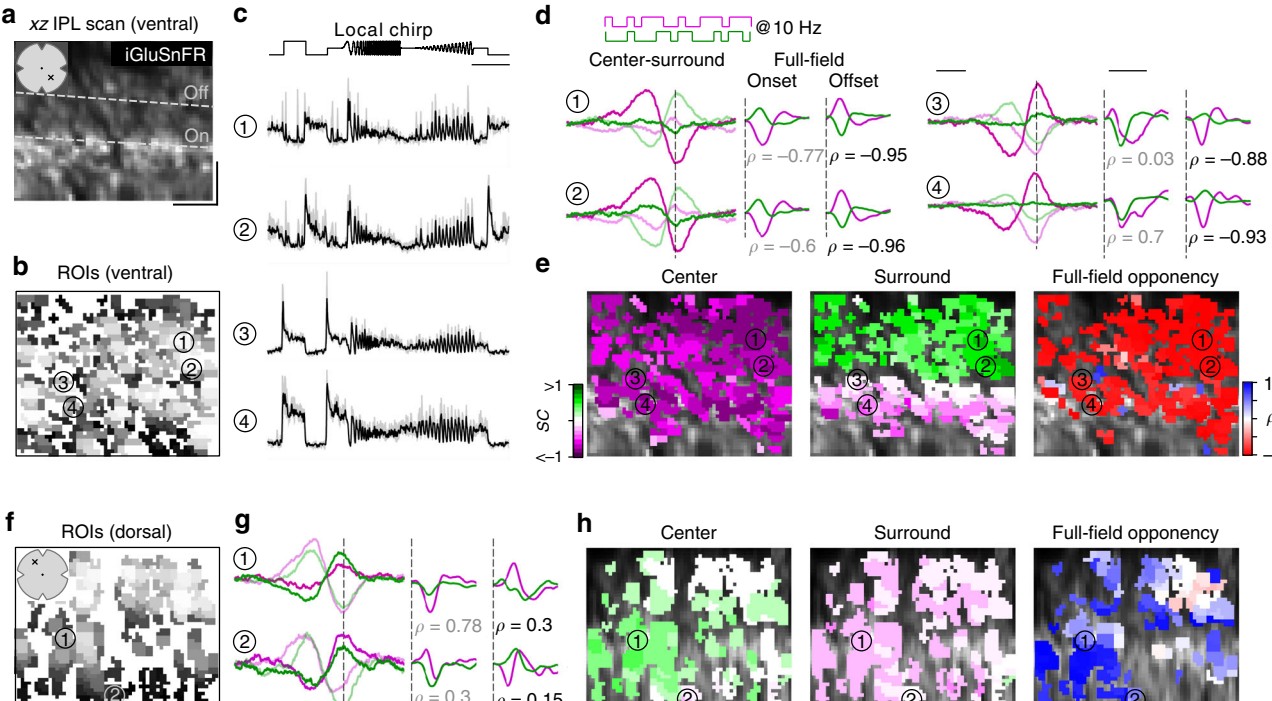

**Fig. 3 Recording chromatic bipolar cell responses across the inner plexiform layer. a** Vertical (xz) scan field (48 × 50 μm, 11.17 Hz) located in the ventral retina showing iGluSnFR expression across the inner plexiform layer (IPL). Dotted lines indicate On and Off ChAT band, respectively. x and y scale bars: 10 μm. **b** ROI mask for scan field shown in **a**. For details on ROI detection, see Methods. **c** Mean glutamate traces (black, s.d. shading in gray; $n = 5$ trials) of ROIs indicated in **b** in response to a local chirp stimulus (scale bar: 5 s). **d** Temporal center (bright) and surround (dim) kernels (event-triggered average) estimated from a 10 Hz center-surround flicker stimulus of UV and green LED (left, scale bar: 200 ms) and mean glutamate events (stimulus-triggered event) in response to onset and offset of a full-field UV and green stimulus (right, scale bar: 0.5 s) for ROIs indicated in **b**. Linear correlation coefficient ($\rho$) of mean glutamate events indicated below traces. For further analysis, $\min(\rho_{onset}, \rho_{offset})$ indicated in black is considered as full-field opponency (Methods). Dotted lines indicate time point of response/stimulus onset/stimulus offset. **e** ROI mask from **b** color-coded according to center (left) and surround spectral contrast (SC) (middle) as well as full-field opponency (right). **f** ROI mask of a scan field located in the dorsal retina. **g**, **h** Like **d**, **e**, but for dorsal scan field shown in **f**. Scan fields/traces shown in this figure correspond to representative examples. In total, we recorded $n = 21$ scan fields in $n = 5$ mice. For quantification, see Fig. 4.

To record responses from BCs, we again used ubiquitous expression of iGluSnFR. In contrast to previous work[28], where the scan fields were parallel to the retinal layers, we here employed axial scanning using an electrically tunable lens to rapidly shift the focal plane of the excitation laser[35]. This allowed us to simultaneously record the glutamatergic signals across the entire IPL (Fig. 3a). Like before, we defined ROIs based on local image correlation (Fig. 3b; Methods)[35]. To register the IPL depth of each ROI, we used the two characteristic dendritic plexi of cholinergic starburst amacrine cells as landmarks (cf. Fig. 1b in ref. [28]); these ChAT bands were visible through their TdTomato-expression in our transgenic animals.

In total, we recorded light-evoked BC glutamate release from 3604 ROIs ($n = 21$ scan fields, $n = 5$ mice) across the entire IPL (Supplementary Fig. 2a). As expected from the type-specific axonal stratification profiles of BCs[34], ROIs located at different IPL depths showed distinct responses to the local chirp stimulus (Fig. 3c; 100 μm in diameter). To investigate chromatic signaling in BCs, we used a 10 Hz center-surround UV and green flicker stimulus (Fig. 3d; Methods). From the glutamate responses of each ROI, we estimated the preferred stimulus (event-triggered stimulus kernels) for the four conditions—center and surround for both UV and green—to obtain the BC ROI's chromatic RF preferences (as SC, see above). In addition, we computed the mean glutamate event of each ROI to the onset and offset of a full-field UV and green light spot (stimulus-triggered event

kernels) to test for full-field color-opponency (quantified by the linear correlation coefficient ($\rho_{onset}, \rho_{offset}$) between UV and green onset and offset event kernels; Methods). For a sensitive measure of color-opponency, full-field opponency was determined as min ($\rho_{onset}, \rho_{offset}$) and thus cells with antagonistic responses to either the onset or offset of a full-field UV and green light spot ($\rho_{onset} < -0.3$ or $\rho_{offset} < -0.3$) were considered as color-opponent.

In-line with the chromatic preference of cone center responses (cf. Fig. 2), we found that BCs located in the ventral and dorsal retina showed a UV- and green-dominant center, respectively (Figs. 3d–h, 4a, b; ventral: $SC_{center} = -0.44 \pm 0.24$, dorsal: $SC_{center} = 0.1 \pm 0.22$). Overall, we did not observe large differences in SC of BC center responses across the IPL (Supplementary Fig. 2b). This is consistent with recent connectomic data demonstrating that, except for type 9 and type 1 BC (Discussion), mouse BCs indiscriminately contact all cone types within their dendritic tree[30].

The chromatic preference of BC surround responses differed from that of their respective center responses, particularly in the ventral retina. Surround responses in ventral BCs were systematically shifted towards green ($SC_{surround} = 0.21 \pm 0.27$), resulting in color-opponent full-field responses for approx. 75% of the ventral BC ROIs (Figs. 3d, e, 4a–d; $n = 1020/1348$). The difference in SC of center and surround ($SC_{Diff}$) was significantly larger for ROIs located in the IPL's Off sublamina compared to those in the On sublamina (Fig. 4e, f, Supplementary Fig. 2c;

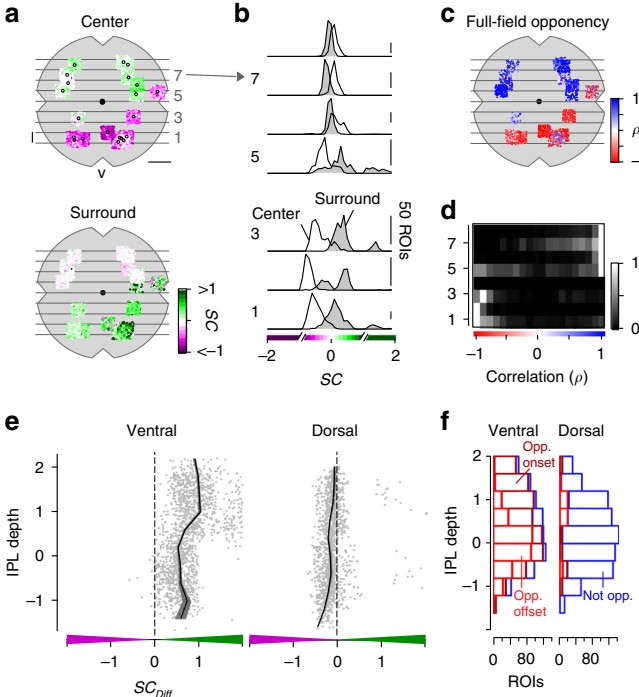

**Fig. 4 Chromatic signals of bipolar cells match cone responses across the retina. a** Distribution of recorded inner plexiform layer (IPL) scan fields (black circles), with ROIs color-coded according to their center (top) and surround spectral contrast (SC; bottom). ROIs are scattered around scan field center by ±300 μm in x and y. Gray lines and numbers on the left indicate bins used for analysis in **b**, **d**. Bin size: 0.5 mm, scale bar: 1 mm. **b** Distribution of center (no fill) and surround (gray fill) SC values from ventral to dorsal retina. Numbers indicate bins shown in **a**. For all bins, center SC was significantly different from surround SC (Linear mixed-effects model for partially paired data; see Methods and Supplementary Methods). **c** Same as **a**, but color-coded according to full-field opponency (min($\rho_{onset}$, $\rho_{offset}$); Methods). **d** Peak-normalized histograms showing distribution of correlation coefficients of onset and offset full-field events from ventral to dorsal retina. Numbers indicate bins shown in **a**. Bin size: 0.5 mm. **e** Difference of center and surround SC ($SC_{Diff}$) across the IPL for ROIs located in the ventral (left) and dorsal (right) retina. $SC_{Diff}$ significantly varied with IPL depth for both ventral and dorsal retina (Generalized additive model; see Methods and Supplementary Methods). **f** Distribution of onset (dark red; $\rho_{onset} < -0.3$) and offset (light red; $\rho_{offset} < -0.3$) full-field opponent and non-opponent (blue) ROIs across the IPL for ventral and dorsal scan fields. Number of opponent ROIs significantly varied with IPL depth for ventral and dorsal retina (Generalized additive model; see Methods and Supplementary Methods).

Discussion). Interestingly, Off BCs showed more color-opponent responses to the onset of a full-field light spot, while On BCs exhibited color-opponency more frequently upon light offset (Discussion). Dorsal BCs showed a shift towards slightly higher UV-sensitivity in their surround responses (Figs. 3g, h, 4a, b; $SC_{surround} = 0.03 \pm 0.19$), which was stronger for On compared to Off BCs (Fig. 4e) but much smaller than for ventral BCs; therefore only very few ($n = 144/1442$) dorsal BCs showed color-opponent full-field responses (Fig. 4f). In addition, we obtained comparable results when modulating green and UV sinusoidally (Supplementary Fig. 3)—a visual stimulus often used in retinal studies on chromatic processing (e.g., refs. [11,25]).

In summary, our data show that BCs provide chromatically tuned excitatory drive to downstream amacrine cell (AC) and RGC circuits. Furthermore, the difference between On and Off

BCs indicates that they might not simply relay the chromatic information from cones to the inner retina.

**Color-opponent responses are preserved at the retinal output.** Finally, we investigated how the chromatic information is represented in the population of RGCs. We used the synthetic calcium indicator Oregon–Green BAPTA-1 (OGB-1) and bulk electroporation to uniformly label the ganglion cell layer (GCL; Fig. 5a)[36,37]. This allowed us to densely record somatic signals from RGCs and displaced ACs (dACs), which make up the mouse GCL at a ratio of roughly 1:1[38]. We recorded GCL scan fields at different positions along the retina's dorso-ventral axis (cf. Fig. 6a). To assign the recorded cells ($n = 8429$ cells, $n = 100$ scan fields, $n = 20$ mice) to functional RGC and dAC groups (presumably corresponding to single types) previously described[37], we presented two achromatic stimuli (full-field chirp and moving bar; Fig. 5b). Like for the BC recordings, we characterized the cells' chromatic preference and full-field opponency by estimating center-surround stimulus and full-field event kernels, respectively, from calcium responses to a 5 Hz center-surround UV and green flicker stimulus (center: 250 μm in diameter) (Fig. 5c).

We found that the chromatic preference of GCL cell center responses largely matched the opsin expression, with a gradient of UV- to green-dominated responses from ventral to dorsal retina (Figs. 5d, h, 6a, b; ventral: $SC_{center} = -0.35 \pm 0.27$, dorsal: $SC_{center} = 0.06 \pm 0.25$). Notably, the chromatic tuning of center responses was more diverse in the GCL compared to the IPL (Supplementary Fig. 4). For example, we frequently observed ventral GCL cells responding stronger to green than to UV center stimulation (Fig. 5c, d), which was not the case for ventral OPL and IPL recordings.

Surround responses of ventral GCL cells were systematically shifted towards green (Figs. 5c, d, 6a, b; $SC_{surround} = 0.21 \pm 0.82$), resulting in a large difference in center vs. surround chromatic preference and, thus, in color-opponent full-field responses ($n = 1312/4247$). For dorsal scan fields, the difference between center and surround chromatic preference and, likewise, the fraction of color-opponent responses was smaller ($SC_{surround} = 0.17 \pm 0.62$; $n = 191/1675$). Interestingly, in our dataset we only rarely observed GCL cells with center-opponent responses (Supplementary Fig. 5; Discussion), which have been found in primates (e.g., ref. [12]) and some dichromatic mammals (e.g., ref. [39]).

In summary, while color-opponency was largely preserved at the level of the retina's output layer, our findings suggest that the complexity of chromatic signals increases from the IPL to the GCL. Next, we investigated whether the color-opponent GCL cells correspond to RGCs and/or dACs and if chromatic information is processed in a type-specific manner.

**Cell-type-specific chromatic processing in retinal ganglion cells.** We next allocated the recorded cells to the previously described functional RGC and dAC groups[37] based on their responses to the achromatic stimuli (Methods). Because color-opponency was pronounced in the ventral retina, we focused the analysis on ventral scan fields. We found that color-opponent GCL cells were assigned to both RGC and dAC groups (Fig. 7a, Supplementary Fig. 6), suggesting that color-opponency is a feature of both cell classes. To verify this, we dye-injected and morphologically reconstructed color-opponent GCL cells ($n = 19$) subsequent to functional imaging (Fig. 7b, Supplementary Fig. 7). Consistent with the abundance of color-opponent responses in the GCL (cf. Fig. 5c), we found a large variety of dendritic morphologies in our sample, with approx. half ($n = 8$) of the reconstructed cells corresponding to dACs, as identified by the absence of an axon. Due to similar response profiles, 5/19

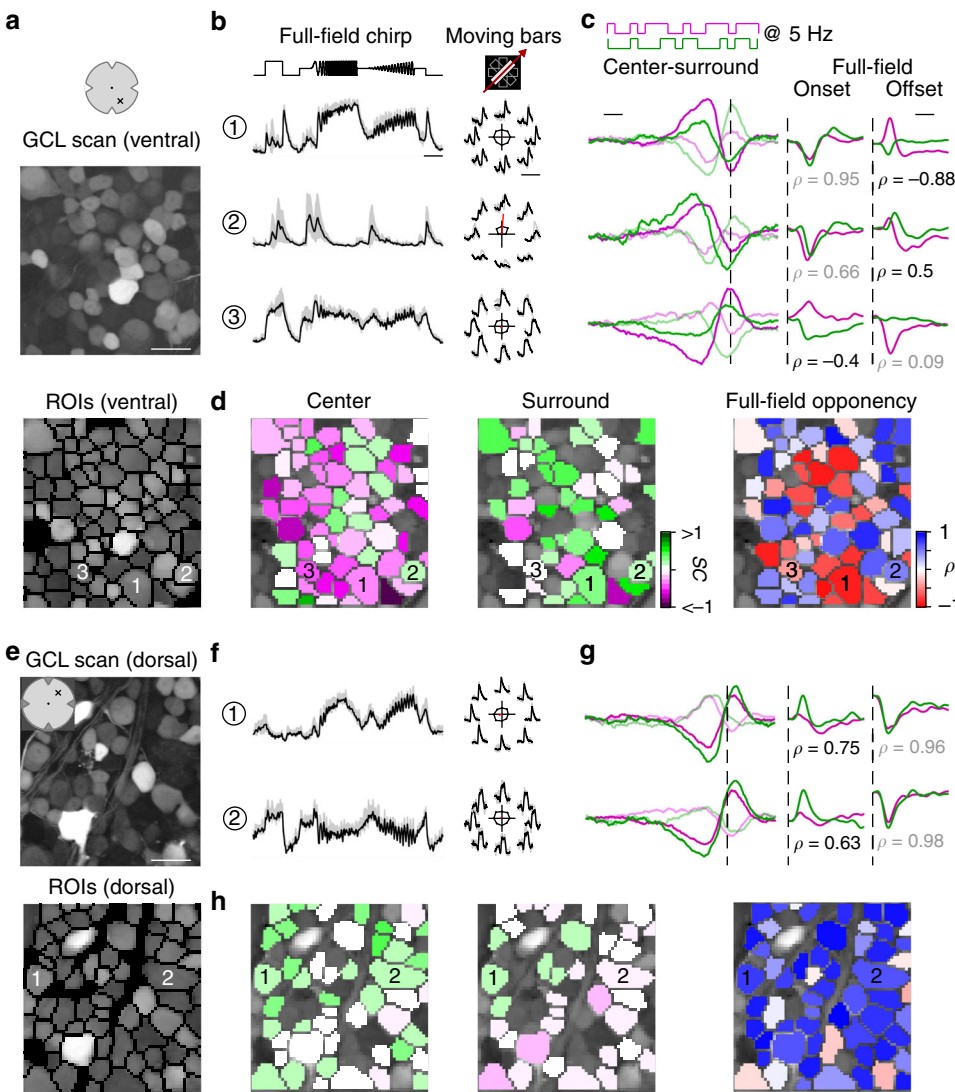

**Fig. 5 Chromatic responses in the ganglion cell layer of the mouse retina. a** Ganglion cell layer (GCL) scan field (top; 95 × 95 μm, 7.8125 Hz) located in the ventral retina electroporated with the synthetic calcium indicator Oregon–Green BAPTA-1 (OGB-1) and corresponding ROI mask (bottom). Scale bar: 20 μm. **b** Mean calcium traces (black, s.d. shading in gray; n = 5 trials) of ROIs indicated in (**a**, bottom) in response to full-field chirp (left, scale bar: 3 s) and moving bars (right, scale bar: 4 s). **c** Temporal center (bright) and surround (dim) kernels estimated from a 5 Hz center-surround flicker stimulus of UV and green LED (left, scale bar: 200 ms) and mean calcium events in response to onset and offset of a full-field UV and green stimulus (right, scale bar: 0.2 s) for ROIs indicated in (**a**, bottom). Linear correlation coefficient of full-field events indicated below traces. For further analysis, min($\rho_{onset}$, $\rho_{offset}$) indicated in black is considered as full-field opponency (Methods). Dotted lines indicate time point of response/stimulus. **d** ROIs from (**a**, bottom) color-coded according to center (left) and surround spectral contrast (SC; middle) as well as full-field opponency (right). **e** Scan field and corresponding ROI mask located in the dorsal retina. Scale bar: 20 μm. **f–h** Like **b–d**, but for scan field shown in **e**. Scan fields/traces shown in this figure correspond to representative examples. In total, we recorded n = 100 scan fields in n = 20 mice. For quantification, see Figs. 6, 7.

morphologically identified RGCs and dACs were assigned to the wrong class (Supplementary Fig. 7). Because we were interested in chromatic retinal output, in the following we focused on RGCs (for dACs, see Supplementary Fig. 6).

Ventral color-opponent RGCs were assigned to diverse functional groups, including Off, On-Off, and On groups (Fig. 7a). Most RGC groups (27/32) contained at least a few (n ≥ 3) color-opponent cells, indicating that color-opponency may partially be inherited from BCs (cf. Fig. 4). Surprisingly, the fraction of color-opponent cells greatly varied across groups: Most color-opponent RGCs were assigned to groups displaying sustained On or Off responses. For example, many sustained On alpha cells ($G_{24}$) showed color-opponent responses (Fig. 7b), consistent with an

earlier study[25]. In addition, color-opponency was a prominent feature in $G_{27}$, which exhibited a bistratified morphology (Fig. 7c, Supplementary Fig. 7), reminiscent of RGC-type 73 in ref. [40], and $G_{31}$, which corresponds to an Off contrast-suppressed type. Other than in BCs, where color-opponency was a prominent feature across the complete population of BC types spanning the IPL, most transient On and some On-Off and Off RGC groups contained relatively few color-opponent cells. This difference between BCs and RGCs suggests that RGC color-opponency is not just only inherited from BCs (Discussion).

The most parsimonious explanation for such RGC-type-dependent differences in fraction of color-opponent cells is that groups differ in their center and surround spectral preference,

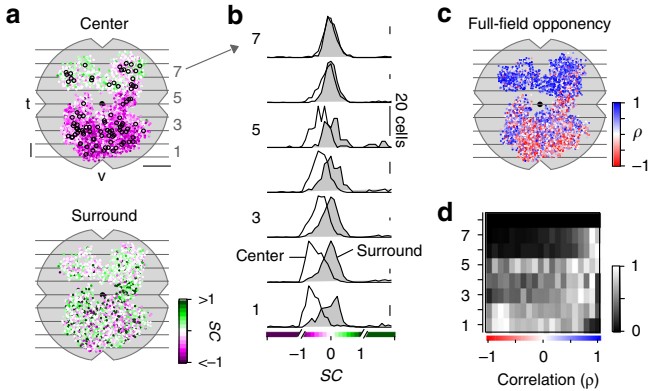

**Fig. 6 Color-opponency at the level of the mouse retinal output. a** Distribution of recorded ganglion cell layer (GCL) scan fields (black circles), with ROIs color-coded according to their center (top) and surround spectral contrast (SC; bottom). ROIs are scattered around scan field center by ±300 μm in x and y. Gray lines and numbers on the left indicate bins used for analysis in **b**, **d**. Bin size: 0.5 mm, scale bar: 1 mm. **b** Distribution of center (no fill) and surround (gray fill) SC values from ventral to dorsal retina. Numbers indicate bins shown in **a**. For all bins, center SC was significantly different from surround SC (Linear mixed-effects model for partially paired data; see Methods and Supplementary Methods). **c** Same as **a**, but color-coded according to full-field opponency (min($\rho_{onset}$, $\rho_{offset}$); Methods). **d** Peak-normalized histograms showing distribution of correlation coefficient of full-field opponency from ventral to dorsal retina. Numbers indicate bins shown in **a**. Bin size: 0.5 mm.

with a larger difference between these two preferences resulting in more color-opponent cells. Additionally, nonlinear integration of center and surround chromatic information could lead to pronounced color-opponency in specific RGC groups. To distinguish between these two possibilities, we tested how well the percentage of color-opponent cells within a group was explained by its chromatic preference using a permutation test: For each group with >15 assigned cells (29/32), we generated a distribution of expected percentages of color-opponent cells—given the cells' center and surround preference, but shuffling their group labels—and compared it to the observed percentage of color-opponent cells (Fig. 7d; Methods). We found that in ~55% (16/29) of all RGC groups investigated, the number of color-opponent cells was explained by the difference in chromatic preference between center and surround. However, the remaining groups showed either a significantly lower or higher percentage of color-opponent cells than expected, indicative of nonlinear integration of center and surround chromatic responses in these groups. The seven groups with fewer color-opponent cells ($G_2$, $G_{17}$, $G_{18}$, $G_{20}$, $G_{21}$, $G_{25}$, and $G_{32}$) comprised a heterogeneous set of RGC groups, including Off, transient On and contrast-suppressed ones (Fig. 7d). In contrast, four of the six groups with higher percentages of color-opponent cells than expected all showed slow On responses ($G_{22}$, $G_{26}$, $G_{27}$, and $G_{28}$) and the other two displayed sustained Off responses ($G_7$ and $G_{31}$). Interestingly, three of the four dAC groups with significantly more color-opponent cells than expected also showed slow On responses (Supplementary Fig. 6), which might hint at a common circuit mechanism.

In summary, our data showed that color-opponency is a widespread feature among ventral RGC groups that is partially inherited by presynaptic BC circuits. However, we found evidence for nonlinear integration of chromatic information in a subset of RGC groups, increasing the diversity of chromatic responses at the level of the retinal output.

## Discussion

In this study, we systematically surveyed chromatic signaling across three consecutive processing stages in the mouse retina by population imaging of the chromatic output signals of cones, BCs and RGCs. We showed how color-opponency present in the ventral retina is already created at the cone synapse by lateral inhibition from HCs that is at least partially driven by rod photoreceptors. In addition, we demonstrated that BCs then relay the chromatic information to RGCs in the inner retina, where type-specific, nonlinear center-surround interactions result in specific color-opponent output channels to the brain. Our finding that color-opponency is mostly limited to the S-opsin dominant ventral retina is consistent with behavioral experiments suggesting that color vision in mice may be largely restricted to their upper visual field[23].

Many vertebrate species show selective wiring between distinct types of cones and HCs, which generates color-opponent responses already in the outer retina (reviewed in ref. [6]). Similarly, it has recently been demonstrated that color-opponency in Drosophila is already present at the level of the photoreceptor output[41], generated by an evolutionary convergent center-surround mechanism involving HC-like inhibitory interneurons[42]. In the primate retina, two types of HC that preferentially contact S- or L/M-cones provide a chromatically opponent antagonistic surround to cones[7,43]. In contrast, mice and some other mammalian species only possess one HC type[44]. As it indiscriminately contacts S- and M-cones, its role in chromatic processing has been much less clear. By recording the glutamatergic output of cones in the intact, whole-mounted retina, we were able to demonstrate that also in mouse color-opponency is already present at the level of the cone output. Specifically, UV-sensitive cones located in the ventral mouse retina exhibited green-sensitive surround responses, mediated by rod signals that are relayed to cones via HCs. This is consistent with a recent study showing that color-opponent responses of ventral JAM-B RGCs originate from a rod-cone opponent mechanism involving HCs[26]. As HCs form highly stereotypical connections with cones and rods (e.g., ref. [45]), it is likely that rods also contribute in a similar way to the surround of cones located in the dorsal retina. However, since most dorsal cones express green-sensitive M-opsin with a peak sensitivity close to that of rhodopsin (cf. Fig. 1a), a surround mediated by rods would not result in color-opponency of these neurons. The prerequisites for such a rod-mediated mechanism have been experimentally established: First, rod signals travel in HCs from the axon terminals to the soma via the HC axon (ref. [46]; but see ref. [47]). Second, mouse rods can drive visual responses at the low photopic light levels generated by our visual stimulation (and laser-induced background activity; for discussion, see Supplementary Figs. 9, 10 and Methods), likely by a combination of mechanisms involving both outer[48] and inner[49] segments. This has been demonstrated across different levels of the visual system, ranging from rod-driven BC[28,50] and RGC responses[51] in the isolated retina (without intact pigment epithelium) to rod-driven visually guided behavior[52–54].

Depending on the mechanism used, retinal circuits extracting chromatic information can be roughly classified into two categories: cone-type-selective and cone-type-unselective. Usually, cone-type-selective retinal circuits depend on BC types that preferentially sample from specific spectral cone types. The best described example for such a cone-type-selective pathway is likely the circuit generating blue–yellow opponency in the primate retina. Here, the so-called small bistratified RGC receives blue-On and yellow-Off input from BCs that exclusively contact and largely avoid S-cones, respectively[11,12]. Cone-type-selective BCs have also been identified in most dichromatic mammals

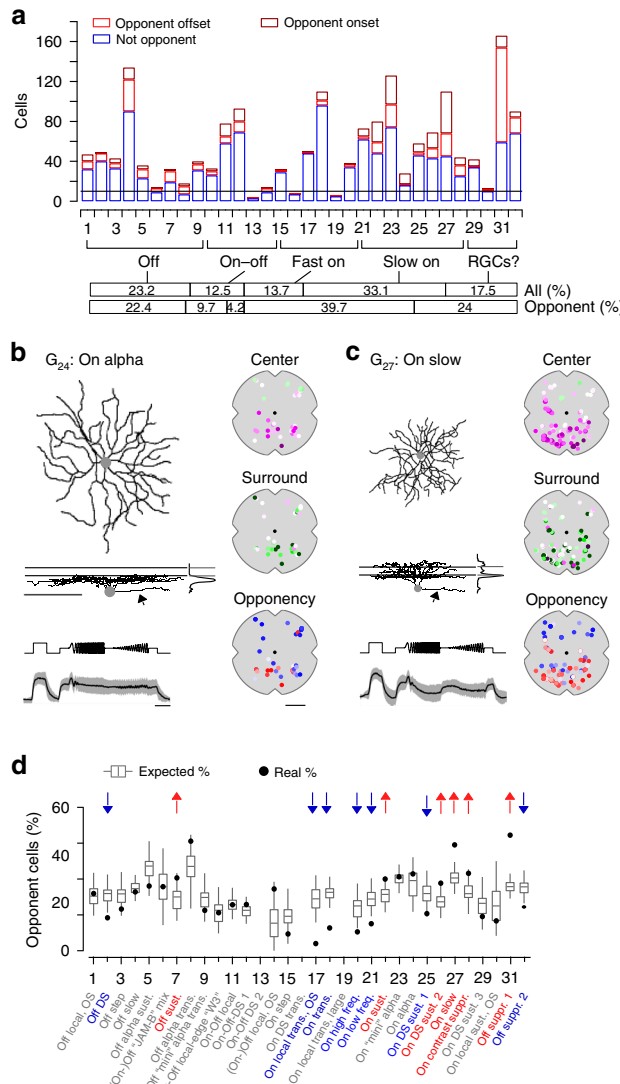

**Fig. 7 The color-opponent output channels of the mouse retina. a** Distribution of onset (dark red; $\rho_{onset} < -0.3$) and offset (light red; $\rho_{offset} < -0.3$) color-opponent and non-opponent (blue) RGCs located in the ventral retina. For analysis in **d**, onset and offset color-opponent cells were summed and only groups with $n > 10$ cells were used. Bars in the bottom illustrate the frequency (in percent) of different functional RGC classes (Off, On-Off, Fast On, Slow On and Uncertain RGCs; based on ref. [37]), considering all RGCs (black) and color-opponent RGCs (onset and offset), respectively, located in the ventral retina. **b** Dendritic morphology with stratification profile of an exemplary RGC assigned to group $G_{24}$, dye-filled and reconstructed subsequent to imaging experiments. Lines for side-view of morphology and stratification profile indicate On and Off ChAT bands. Arrows point at axon present only for RGCs and not for dACs. Scale bar: 100 μm. Bottom shows mean full-field chirp response (black, s.d. shading in gray; $n = 5$ trials; scale bar: 3 s) of RGC group $G_{24}$. Panels on the right show recording positions of all cells assigned to $G_{24}$, color-coded according to center (top) and surround (middle) spectral contrast (SC) and full-field opponency (bottom). Scale bar: 1 mm. **c** Like **b**, but for $G_{27}$. **d** Box plots (quartile method: Turkey; whisker method: s.d.) show distribution of expected percentages of color-opponent cells given center SC and $SC_{Diff}$ values in each group (for details, see Methods). Black circles indicate true percentage of color-opponent cells. Arrows pointing down- and up indicate groups with significantly more and less color-opponent cells than expected, respectively, (for cell numbers per group, see **a**). $G_1$:0.43, $G_2$:0.006, $G_3$:0.1, $G_4$:0.27, $G_5$:0.1, $G_6$:0.44, $G_7$:0.033, $G_8$:0.052, $G_9$:0.19, $G_{10}$:0.46, $G_{11}$:0.47, $G_{12}$:0.14, $G_{13}$:-, $G_{14}$:0.053, $G_{15}$:0.1, $G_{16}$:-, $G_{17}$:0, $G_{18}$:0, $G_{19}$:-, $G_{20}$:0.019, $G_{21}$:0.0014, $G_{22}$:0.021, $G_{23}$:0.42, $G_{24}$:0.33, $G_{25}$:0.025 $G_{26}$:0.03 $G_{27}$:0, $G_{28}$:0.028, $G_{29}$:0.15, $G_{30}$:0.33, $G_{31}$:0, $G_{32}$:0.0061. A permutation test was used to assess significance (see Methods).

(reviewed in ref. [14]). For example, mice possess an On BC type exclusively contacting S-cones (type 9) and an Off BC type that prefers M-cones (type 1)[21,30,55]. Therefore, in dorsal scan fields we expected to find UV-dominant center responses in the innermost IPL layer, where S-cone selective type 9 BCs stratify[21,30]. However, we found such responses only rarely. The low frequency of presumed type 9 BC responses resonates well with their very sparse axonal arbors. Based on EM data we estimated to find ~1 ribbon synapse per IPL scan field (cf. Fig. 4f in ref. [35]). In addition, we did not observe a bias for purely green center responses in the Off sublamina of dorsal scan fields, as would be expected for M-cone preferring type 1 BCs[30,55]. This may be explained by a relatively small difference in chromatic preference of type 1 compared to other BC types (cf. Fig. 6 in ref. [55]).

In primates, cone-type-selective BCs provide separate chromatic input channels to RGCs, generating a center-opponent RF structure[11,12]. In dichromatic mammals, a similar mechanism results in center-opponent RGCs in ground squirrel[39] and likely guinea pig[56] and rabbit[57]. Such a circuit could also exist in mice—at least in the dorsal retina where opsin co-expression is low. However, center-opponent RGC RFs were rare in our dataset and did not comprise of a single functional type. For identifying center opponency, the stimulus should ideally be aligned to the RF center of the recorded cell. However, this is not possible in our

population approach, where the stimulus is aligned to the center of each recording field (Methods), resulting in a spatial offset of up to 50 μm between stimulus and RF center of the recorded cell. Therefore, we might have underestimated the number of center-opponent RGCs. Until now, there is evidence for only one mouse RGC type that uses cone-type-selective BC input: It features a UV-dominant center and a green surround, with the former generated by a bias for connecting to type 9 BCs[24]. Therefore, connectivity matrices (e.g., refs. [30,34]) obtained from large-scale EM reconstructions may result in further candidate cone-type-selective pathways.

Red–green opponency in the primate retina is an example of a cone-type-unselective circuit. It is a consequence of midget RGCs receiving input from one or very few M- and L-cones, resulting in either green- or red-dominant center RFs, that are compared to a yellow (M + L) surround[7–9]. Similarly, two color-opponent pathways independent of cone-type-selective connectivity have been identified in the mouse retina. First, the asymmetric opsin distribution can result in color-opponent responses of RGCs located along the horizontal midline due to chromatically distinct input to their center and surround[25]. Second, a rod-cone opponent pathway has been linked to color-opponency in JAM-B RGCs located in the S-opsin dominated ventral retina[26]. Our results suggest that the latter mechanism is not restricted to a single RGC type, but that most color-opponent responses in the mouse retina are inherited from the outer retina. Specifically, retinal neurons in the S-opsin dominated ventral retina will display color-opponent responses if the ratio of rod/cone input differs for their center and surround—similar to red–green opponency in primates (e.g., ref. [58]) and color-opponent RGCs in the opsin transitional zone of mice[25]. Importantly, this mechanism does not rely on pathways that avoid rods and are selective for cones. For example, signal spread from rods to cones via gap junctions (e.g., ref. [59]) or direct rod contacts to Off BCs

(e.g., ref. [30]) will reduce the difference in rod/cone input ratio of center and surround. Nevertheless, color-opponency might still be a prominent feature of these neurons—similar to midget RGCs in the peripheral retina with only a slight bias towards L- or M-cone input in their center compared to their surround[58].

In line with this, we found that the complete population of ventral BCs conveyed chromatic information to downstream circuits. Interestingly, the difference in center and surround chromatic preference was larger for Off compared to On BCs. The BCs' inhibitory surround could originate from HCs and/or GABAergic wide-field ACs in the outer and inner retina, respectively. We found that in the ventral retina the surround mediated by HCs is largely green-sensitive. In contrast, the surround mediated by ACs is likely UV-sensitive, as wide-field ACs receive their excitatory drive from BC center responses (e.g., ref. [60]), which, in the ventral retina, are UV-dominant. Therefore, the more pronounced difference in center and surround chromatic preference in Off BCs may be due to a stronger contribution of HCs compared to ACs in generating the Off BCs' inhibitory surround. Despite this difference, full-field color-opponency was a prominent feature of both On and Off BCs. That On and Off BCs displayed color-opponent responses more frequently to the offset and onset of a full-field light spot, respectively, suggests that different presynaptic circuits might mediate color-opponency in response to light increment and decrement. Interestingly, these differences between On and Off BCs were not preserved at the level of the retinal output. Here, many transient On and some Off RGC groups contained fewer color-opponent cells than expected from their center and surround chromatic preference. In contrast, color-opponency was significantly enriched in some sustained Off RGC, slow On RGC and slow On dAC groups. This indicates that the excitatory drive from BCs is heavily processed in a RGC-type-specific way, likely by selective connectivity to a set of ACs[34] and type-specific dendritic processing[61], demonstrating how the representation of a single feature like color-opponency changes across subsequent processing layers. However, as the size of the center stimulus used for IPL and GCL recordings was different, explaining exactly how observed RGC responses arise from recorded BC responses requires further investigation.

In summary, our data provide little evidence for cone-type-selective circuits in the mouse retina. Instead, most color-opponent responses originate in the outer retina, likely generated by a rod-cone opponent pathway. In the inner retina, chromatic information from cones is further processed, resulting in type-specific chromatic responses at the level of the retinal output.

The asymmetric opsin distribution divides the mouse retina into distinct regions. The dorsal part resembles the cone mosaic of other dichromatic mammals, with many M-cones and few S-cones[62]. Therefore, one would expect that the evolutionary conserved circuits that extract blue-green opponency (reviewed in ref. [6]) also exist in the dorsal retina of mice. In contrast, the ventral part of the mouse retina, with its M-cones co-expressing S-opsin[19,20], was long considered unfit for color vision. Instead, it was linked to optimal sampling of achromatic contrast information in the sky portion of natural scenes[18]. Here we show that in fact, color-opponent neurons are predominantly located in the ventral retina of mice. This is in agreement with previous RGC studies[25,26] as well as with a recent behavioral study, which demonstrated that mice perform much better in discriminating colored light spots presented in their upper visual field[23]. Using a rod-cone based mechanism to extract chromatic information in the ventral retina may be actually advantageous, because it allows color vision[23] and detecting dark objects such as predatory birds[18] through the widespread expression of S-opsin. This arrangement might also be relevant in other species with a regional increase in S-opsin density in their retina (reviewed in ref. [63]), including the common shrew or some hyenas. Because mouse rod photoreceptors are active in the photopic regime[51–54], rod-cone opponency likely contributes to the animal's color vision across a substantial intensity range, increasing its relevance for informing behavior.

According to the efficient coding theory, sensory systems adapt to the distribution of signals present in their natural environment[64]. That color-opponency of mice appears to be largely restricted to their ventral retina suggests that behaviorally relevant chromatic information should be found in their upper visual field. It has been speculated that mice use color vision for social communication by detecting urine tags[26]. However, urine tags large enough to appear in the upper visual field have so far only been observed for mice housed under unnatural conditions[65]. In addition, urine might not constitute a reliable visual cue under natural conditions[66], especially since mice olfaction would be the more obvious choice to detect and analyze urine cues. Alternatively, as most predators are expected to approach the mouse from above, color vision in the upper visual field could well support threat detection. Especially for visual scenes with inhomogeneous illumination (e.g., forest), that result in large intensity fluctuations at the photoreceptor array, color-opponent RF structures may result in a more reliable signal (discussed in refs. [67,68]).

## Methods

**Animals and tissue preparation.** All animal procedures were approved by the governmental review board (Regierungspräsidium Tübingen, Baden-Württemberg, Konrad-Adenauer-Str. 20, 72072 Tübingen, Germany) and performed according to the laws governing animal experimentation issued by the German Government. For all experiments, we used 5- to 18-week-old mice of either sex. For OPL recordings, we used Cx57$^{+/+}$ ($n = 9$; ref. [69]) mice, which were negative for Cre recombinase on both alleles and, therefore, could be considered wild-type animals. In addition, we used the HR2.1:TN-XL ($n = 3$) mouse line where the calcium indicator TN-XL was exclusively expressed in cones (Supplementary Fig. 8)[70]. For IPL recordings, we used ChAT$^{Cre}$ ($n = 5$, JAX 006410, The Jackson Laboratory; ref. [71]) mice and for GCL/RGC recordings we used C57Bl/6 J ($n = 14$, JAX 000664) or Pvalb$^{Cre}$ ($n = 9$, JAX 008069; ref. [72]) mice. The transgenic lines ChAT$^{Cre}$ and Pvalb$^{Cre}$ were crossbred with the Cre-dependent red fluorescent reporter line Ai9$^{tdTomato}$ (JAX 007905). Owing to the exploratory nature of our study, we did not use randomization and blinding. No statistical methods were used to predetermine sample size.

Animals were housed under a standard 12 h day/night rhythm at 22° and 55% humidity. For activity recordings, animals were dark-adapted for ≥1 h, then anaesthetized with isoflurane (Baxter) and killed by cervical dislocation. The eyes were enucleated and hemisected in carboxygenated (95% O$_2$, 5% CO$_2$) artificial cerebrospinal fluid (ACSF) solution containing (in mM): 125 NaCl, 2.5 KCl, 2 CaCl$_2$, 1 MgCl$_2$, 1.25 NaH$_2$PO$_4$, 26 NaHCO$_3$, 20 glucose, and 0.5 L-glutamine (pH 7.4). Then, the tissue was either electroporated (see below) or moved to the recording chamber, where it was continuously perfused with carboxygenated ACSF at ~36 °C. In all experiments, ACSF contained ~0.1 μM Sulforhodamine-101 (SR101, Invitrogen) to reveal blood vessels and any damaged cells in the red fluorescence channel[73]. All procedures were carried out under very dim red (>650 nm) light.

**Bulk electroporation.** For recordings in the ganglion cell layer (GCL), the fluorescent calcium indicator Oregon–Green BAPTA-1 (OGB-1) was bulk electroporated[36,37]. In brief, the retina was dissected from the eyecup, flat-mounted onto an Anodisc (#13, 0.2 μm pore size, GE Healthcare) with the GCL facing up, and placed between two 4-mm horizontal plate electrodes (CUY700P4E/L, Nepagene/Xceltis). A 10 μl drop of 5 mM OGB-1 (hexapotassium salt; Life Technologies) in ACSF was suspended from the upper electrode and lowered onto the retina. After application of nine pulses (~9.2 V, 100 ms pulse width, at 1 Hz) from a pulse generator/wide-band amplifier combination (TGP110 and WA301, Thurlby handar/Farnell), the tissue was moved to the recording chamber of the microscope and left to recover for ~30 min before the recordings started.

**Virus injection.** The viral construct AAV2.7m8.hSyn.iGluSnFR was generated in the Dalkara lab (for details, see ref. [74]). The iGluSnFR plasmid construct was provided by J. Marvin and L. Looger (Janelia Research Campus, USA). A volume of 1 μl of the viral construct was then injected into the vitreous humour of 3- to 8-week-old mice anaesthetized with 10% ketamine (Bela-Pharm GmbH & Co. KG) and 2% xylazine (Rompun, Bayer Vital GmbH) in 0.9% NaCl (Fresenius). For the

injections, we used a micromanipulator (World Precision Instruments) and a Hamilton injection system (syringe: 7634-01, needles: 207434, point style 3, length 51 mm, Hamilton Messtechnik GmbH). Imaging experiments were performed 3–4 weeks after injection. As iGluSnFR expression tended to be weaker in the central retina, most OPL and IPL scan fields were acquired in the medial to peripheral ventral or dorsal retina.

**Two-photon imaging.** We used a MOM-type two-photon microscope (designed by W. Denk, MPI, Heidelberg; purchased from Sutter Instruments/Science Products)[73]. In brief, the system was equipped with a mode-locked Ti:Sapphire laser (MaiTai-HP DeepSee, Newport Spectra-Physics), two fluorescence detection channels for iGluSnFR/OGB-1 (HQ 510/84, AHF/Chroma) and SR101/tdTomato (HQ 630/60, AHF), and a water immersion objective (W Plan-Apochromat ×20 /1.0 DIC M27, Zeiss). The laser was tuned to 927 nm for imaging iGluSnFR, OGB-1 or SR101, and to 1000 nm for imaging tdTomato. For image acquisition, we used custom-made software (ScanM by M. Müller and T.E.) running under IGOR Pro 6.3 for Windows (Wavemetrics), taking time-lapsed $64 \times 64$ pixel image scans (at 7.8125 Hz) for OGB-1 imaging in the GCL and $128 \times 128$ pixel image scans (at 3.9 Hz) for glutamate imaging in the outer plexiform layer (OPL). For vertical glutamate imaging in the IPL, we recorded time-lapsed $64 \times 56$ pixel image scans (at 11.16 Hz) using an electrically tunable lens (for details, see ref. [35]). For high resolution images, $512 \times 512$ pixel images were acquired. The positions of the fields relative to the optic nerve were routinely recorded.

**Laser-induced effects on retinal activity.** The laser power used in the experiments largely depended on how deep in the retina the imaging plane was located and ranged between 7 and 13 mW. This means that we used lower laser powers for GCL scans (~7 mW) compared to IPL (~9 mW) and OPL scans (~12 mW). The somewhat higher laser powers for IPL and OPL scans were required to compensate for laser power loss due to scattering and absorption within the tissue. Additional adjustments of the laser power to compensate for differences in dye labeling or indicator expression were typically within ±1 mW. Two-photon imaging introduces a constant laser-induced baseline activity (see below and refs. [73,75]). To test the effect of the two-photon laser on chromatic RGC responses, we electrically recorded from single RGCs with and without laser exposure, while displaying center-surround flash and flicker stimuli used in the imaging experiments (Supplementary Fig. 9). While we found that the green center component of RGC RFs was slightly reduced by the laser-mediated background illumination for flicker stimuli, there was no systematic effect of the two-photon laser on RGC flash responses, independent on the focal plane of the laser. In addition, we investigated the effect of the two-photon laser on rod-mediated surround responses of BCs. For that, we recorded BC glutamate release in a small horizontal IPL plane (~30 × 30 μm) in the ventral retina while increasing the size of the laser-illuminated area, until covered nearly the complete surround stimulus (Supplementary Fig. 10). For our standard IPL recordings, the scan field only partially covers the center stimulus (100 μm in diameter). If the 2 P laser had saturated the rods, we would have expected to see a strong decrease in green surround responses upon extending the laser-illuminated area into the cells' surround. However, we found that changing the laser-illuminated area had little effect on BC activity, including green surround responses. Together, this suggests that for our experimental conditions adaptation of rods and cones to the two-photon laser had only little effect on our results with regard to chromatic processing of retinal neurons.

**Light stimulation.** For light stimulation, we used two different systems. The first system focused a DLP projector (lightcrafter (LCr), DPM-E4500UVBGMKII, EKB Technologies Ltd) with internal UV and green light-emitting diodes (LEDs) through the objective (TTO). To optimize spectral separation of mouse M- and S-opsins, LEDs were band-pass filtered (390/576 Dualband, F59-003, AHF/Chroma). The second system used an LCr with a lightguide port (DPM-FE4500MKIIF) to couple in external, band-pass filtered UV and green LEDs (green: 576 BP 10, F37-576; UV: 387 BP 11, F39-389; both AHF/Chroma), focused through the condenser (TTC) of the microscope (for details, see ref. [76]). For glutamate recordings in the IPL, we solely used the TTO stimulator, while for OPL and GCL recordings we used both TTO and TTC. LEDs were synchronized with the microscope's scan retrace. Stimulator intensity (as photoisomerization rate, $10^3$ P* per s per cone) was calibrated to range from ~0.5 (black image) to ~20 for M- and S-opsins, respectively (for details, see ref. [76]). In addition, a steady illumination component of ~$10^4$ P* per s per cone was present during the recordings because of two-photon excitation of photopigments (discussed in refs. [18,73,75]). The light stimulus was centered to the recording field before every experiment. For all experiments, the tissue was kept at a constant mean stimulator intensity level for at least 15 s after the laser scanning started and before light stimuli were presented.

Two types of light stimuli were used for glutamate imaging in the OPL:
(a) full-field (700 μm in diameter) UV and green flashes,
(b) center (150 μm in diameter) and surround (annulus; full-field flashes sparing the central 150 μm) UV and green flashes.
Three types of stimuli were used for glutamate imaging in the IPL:
(c) local (100 μm in diameter) chirp (for details, see ref. [28]);

(d) 2 Hz sine-wave modulation of center and surround for UV and green LED; and
(e) a UV and green center-surround flicker stimulus, with intensity of center and surround determined independently by a balanced 180-s random sequence at 10 Hz.
Three types of stimuli were used for calcium imaging in the GCL:
(f) full-field (700 μm in diameter) chirp stimulus (for details, see ref. [37]);
(g) 0.3 × 1 mm bright bar moving at 1 mm s$^{-1}$ in eight directions[36]; and
(h) a UV and green center-surround flicker stimulus (250 μm in diameter for center), with intensity of center and surround determined independently by a balanced 300-s random sequence at 5 Hz.
For recording calcium responses in HR2.1:TN-XL mice, we used full-field white flashes (2 s, 50% duty cycle). Sizes of center stimuli were selected to completely fill the scan field area of the recordings and, therefore, did not correspond to RF center sizes of retinal neurons.

**Pharmacology.** All drugs were bath applied for at least 10 min before recordings. The following drug concentrations were used: 50 μM 6,7-dinitroquinoxaline-2,3-dione (NBQX), ACSF with twice the normal concentration of KCl (5 mM). Drug solutions were carboxygenated and warmed to ~36 °C before application.

**Single-cell electrophysiology.** GCL cells were targeted using an infrared LED and CCD camera for intracellular recordings. Electrodes were pulled on a P-1000 micropipette puller (Sutter Instruments) with resistances of 7–15 MΩ and filled with solution consisting of (in mM): 120 K-gluconate, 5 NaCl, 10 KCl, 1 MgCl2, 1 EGTA, 10 HEPES, 2 Mg-ATP, and 0.5 Tris-GTP, adjusted to pH 7.2 using 1 M KOH. Data were acquired using an Axopatch 200B amplifier in combination with a Digidata 1440 (both: Molecular Devices), digitized at 20 kHz and analyzed offline using Igor Pro (Wavemetrics). For recordings, we targeted GCL cells located in the medial retina allowing to investigate the effect of the two-photon laser on both UV and green responses.

**Single-cell injection and morphology reconstruction.** OGB-1-labelled GCL cells were targeted with electrodes (5–15 MΩ) subsequent to two-photon recordings. Single cells in the GCL were dye-filled with SR101 (Invitrogen) using the buzz function (100-ms pulse) of the MultiClamp 700B software (Molecular Devices). Pipettes were carefully retracted as soon as the cell began to fill. Approximately 20 min were allowed for the dye to diffuse throughout the cell before imaging started. After recording, an image stack was acquired to document the cell's morphology, which was then traced semi-automatically using the Simple Neurite Tracer plugin implemented in Fiji. In cases of any warping of the IPL we used the original image stack to correct the traced cells using custom-written scripts in IGOR Pro (for details, see ref. [37]).

**Data analysis.** Data analysis was performed using IGOR Pro.
For GCL recordings, ROIs were defined semi-automatically by custom software[37]. For glutamate imaging in OPL and IPL, ROIs were defined automatically by custom correlation-based algorithms in IGOR Pro. Here, ROI sizes were restricted to match the sizes of cone (3–7 μm diameter) and BC axon terminals (1–4 μm diameter) in OPL and IPL scans, respectively. For OPL recordings, a specific correlation threshold for each scan field was manually selected to account for differences in staining and signal-to-noise ratio. For IPL recordings, correlation thresholds were determined automatically and varied across the IPL due to differences in iGluSnFR labeling and laser intensity (for details, see ref. [35]). For every ROI located in the IPL, depth was determined using the shortest distance of ROI center to TdT-labeled ChAT bands and normalized such that 0 and 1 corresponded to On and Off ChAT band, respectively. The resolution of x–z scans—determined as half maximal width of a Gaussian fit to the measured intensity profile of 170-nm fluorescent beads[35]—was ~0.4 μm in the x–y plane and 1.8 μm along the z-axis. Despite the lower resolution along the z-axis, areas of ROIs estimated from x–z and x–y scans were comparable (3.1 ± 1.5 μm$^2$ and 3.0 ± 1.2 μm$^2$ for x–z and x–y scans, respectively).
To relate each ROIs functional properties to its location on the retina, we registered the orientation of the retina for all IPL and GCL recordings and calculated the linear distance of each ROI to the optic nerve. For most OPL recordings, we did not register the retinal orientation. Here, we used the previously described gradient in opsin expression[18,19] to separate dorsal (mean center SC > 0) and ventral (mean center SC < 0) scan fields.
The glutamate or calcium traces for each ROI were extracted (as ΔF/F) using custom analysis code based on the image analysis toolbox SARFIA for IGOR Pro[77] and resampled at 500 Hz. A stimulus time marker embedded in the recorded data served to align the traces relative to the visual stimulus with 2 ms precision. For this, the timing for each ROI was corrected for sub-frame time-offsets related to the scanning.
First, we detrended the traces by high-pass filtering above ~0.1 Hz. For all stimuli except for the center-surround flicker, we then computed the median activity $r(t)$ across stimulus repetitions ($n = 4–5$ repetitions for chirps, $n = 3$ repetitions for sine, $n = 25–30$ repetitions for full-field and center-surround flashes, $n = 3$ repetitions for moving bars).

*Center-surround stimulus and event kernels*: We mapped the stimulus kernels to the center-surround flicker by computing the glutamate/calcium event-triggered average (event-triggered stimulus kernels). To this end, we differentiated the response trace and estimated a response threshold as:

$$\sigma = \frac{\text{Median}(|r(t)|)}{0.6745} \quad (1)$$

We then computed the glutamate/calcium transient-triggered average stimulus, weighting each sample by the amplitude of the transient:

$$\mathbf{F}(x, y, \tau) = \frac{1}{M}\sum_{i=1}^{M} c(t_i)S(x, t_i + \tau) \quad (2)$$

Here, $S(x, t_i + \tau)$ is the stimulus, $\tau$ is the time lag and $M$ is the number of glutamate/calcium events.

Similarly, for estimating the average glutamate/calcium event kernel upon the onset and offset of full-field stimulation, we first identified time points of full-field light increment and decrement in the stimulus trace and then computed the stimulus-triggered average glutamate/calcium event (stimulus-triggered event kernels).

*Response quality indices*: Kernel quality ($Qi_{Kernel}$) was measured by comparing area under the curve ($F_{Area}$) of each response kernel with the respective baseline:

$$Qi_{Kernel} = 1 - \frac{|F_{Area(Baseline)}|}{|F_{Area(kernel)}|}. \quad (3)$$

Event quality ($Qi_{Event}$) was measured by comparing area under the curve ($F_{Area}$) of each event with the respective baseline:

$$Qi_{Event} = 1 - \frac{|F_{Area(Baseline)}|}{|F_{Area(Event)}|}. \quad (4)$$

To measure how well a cell responded to the other stimuli used (chirp, sine modulation, full-field and center-surround flashes, moving bars), we computed the signal-to-noise ratio

$$Q_i = \frac{\text{Var}[(C)_r]_t}{(\text{Var}[C]_t)_r} \quad (5)$$

where $C$ is the $T$ by $R$ response matrix (time samples by stimulus repetitions), while $()_x$ and $\text{Var}[]_x$ denote the mean and variance across the indicated dimension, respectively[28,37].

For further analysis of chromatic responses, we used

(a) ROIs in the OPL if they showed hyperpolarizing center or full-field responses and $Qi_{full-field} > 0.25$ ($n = 2132/2945$) or $Qi_{center-surround} > 0.25$ ($n = 2008/2589$). For analysis of surround responses, only ROIs with an antagonistic response showing an increase in glutamate release with $F_{Area(Surround)} > (|F_{Area(Center)}|/10)$ were considered ($n = 1057/2589$).

(b) ROIs in the IPL if $Qi_{Kernel} > 0.6$ for center UV or green stimulus kernel ($n = 3188/3604$).

(c) ROIs in the GCL if $Qi_{Kernel} > 0.6$ for center UV or green stimulus kernel ($n = 5922/8429$). For group assignment of GCL cells, we in addition only used ROIs with $Qi_{Chirp} > 0.4$ or $Qi_{Bars} > 0.6$ ($n = 4519/8429$). In addition, we excluded scan fields for which less than 50% of all cells passed the above mentioned quality thresholds ($n = 2$ scan fields).

*Spectral contrast*: For estimating the chromatic preference of recorded cells, we computed a spectral contrast (SC) using the area under the curve ($F_{Area}$) of the mean glutamate traces (OPL recordings; cf. Fig. 1e, g) or the center-surround stimulus kernels (IPL and GCL recordings; cf. Figs. 3d, 5c). For stimulus kernels of IPL and GCL ROIs, we first estimated absolute $F_{Area}$ of each of the four conditions (UV and green center and surround) and then set $F_{Area}$ estimated from kernels anticorrelated to the center kernel to negative values (e.g., antagonistic surround will have negative $F_{Area}$).

Previously, SC has been estimated as Michelson contrast based on dendritic calcium signals in mouse HCs[29], ranging from $-1$ to 1 for the cell responding solely to UV and green, respectively. However, this requires UV and green responses to have the same response polarity (e.g., only decreases in calcium to full-field responses[29]). As both center and surround responses of cells in OPL, IPL, and GCL recordings can have different response polarities to UV and green, we here distinguished three cases to estimate SC.

If green and UV responses had the same response polarity (e.g., cone (1) in Fig. 1e), SC was estimated as Michelson contrast:

$$SC = \frac{|F_{Area(Green)}| - |F_{Area(UV)}|}{|F_{Area(Green)}| + |F_{Area(UV)}|} \quad (6)$$

If the green response had an expected response polarity (e.g., increase in glutamate release upon surround stimulation in cones; cone (2) in Fig. 1l) and the

UV response was antagonistic, SC was estimated as

$$SC = 1 + \frac{|F_{Area(UV)}|}{|F_{Area(Green)}|} \quad (7)$$

Similarly, if the UV response had an expected response polarity and the green response was antagonistic, SC was estimated according to

$$SC = 1 - \frac{|F_{Area(Green)}|}{|F_{Area(UV)}|}. \quad (8)$$

For estimating the difference in SC between center and surround ($SC_{Diff}$), we used:

$$SC_{Diff} = SC_{Surround} - SC_{Center} \quad (9)$$

*Density recovery profiles*: To estimate density recovery profiles (DRPs; ref. [78]) of OPL ROI masks, we first calculated the distance of each ROI to each other ROI within the scan field, binned the distances (bin size = 2 μm) and normalized each bin count to the bin area. Next, we estimated the mean DRP per scan field by averaging the histograms of all ROIs within a field ($n = 56 \pm 30$ ROIs per scan field). To obtain the mean DRPs of all ROI masks, we used $n = 52$ scan fields.

For relating DRPs of the ROIs to anatomy, we used a recent EM dataset of reconstructed cone and rod terminals to estimate anatomical DRPs as described above (cf. Supplementary Fig. 1; $n = 163/2095$ cone/rod terminals; ref. [30]). For calculating a cone DRP with 3% rods (cf. Supplementary Fig. 1d), we first calculated the density of rod terminals and then randomly placed 3% of the expected number of rod terminals across the reconstructed area.

*Field entropy*: Field entropy ($S_{Field}$) was used to estimate the variability of chromatic tuning within single IPL and GCL scan fields (Supplementary Fig. 4). First, SC values of all ROIs within one scan field were binned (bin size: 0.2) and then $S_{Field}$ was defined as

$$S_{Field} = -\Sigma_i p_i \log_2 p_i \quad (10)$$

where $i$ is the number of SC bins and $p_i$ corresponds to the number of ROIs in the $i$th bin. $S_{Field} = 0$ if all ROIs of one recording field are in the same SC bin and therefore have the same spectral tuning. $S_{Field}$ increases if ROIs are equally distributed across multiple bins. In general, high field-entropy indicates high chromatic tuning heterogeneity within a single field. As the number of ROIs per scan field was larger for IPL than GCL recordings, we likely underestimated the difference in $S_{Field}$ between IPL than GCL recordings.

*Full-field opponency*: To measure whether UV and green full-field responses were color-opponent, we calculated the linear correlation coefficient between UV and green event kernels ($Qi_{Event} > 0.25$) to onset and offset of a full-field light spot ($\rho_{onset}$, $\rho_{offset}$). For a sensitive measure of color-opponency, full-field opponency was determined as min($\rho_{onset}$, $\rho_{offset}$) and cells with antagonistic responses to either the onset or offset of a full-field UV and green light spot were considered as color-opponent ($\rho_{onset} < -0.3$ or $\rho_{offset} < -0.3$).

*Sine data*: To estimate the chromatic preference of a cell based on its response to sinusoidal modulation (Supplementary Fig. 3), we first quantified the response phase for every stimulus condition (UV and green center and surround). For every ROI, we cross-correlated the mean glutamate trace of each condition with the stimulus trace and converted the time shift of maximal correlation into degrees. We then extracted the amplitude of the fundamental response component (F1) from the mean glutamate traces using Fourier transform. For the polar plot, response phases of different ROIs were binned using a bin size of 15° and each polar histogram was normalized according to its mean F1 amplitude. We performed this analysis for ventral and dorsal On (IPL depth < 0.2) and Off (IPL depth > 0.5) ROIs separately.

*Direction selectivity*: In order to compute the direction selectivity (DS) of recorded GCL, we first performed singular value decomposition (SVD) on the mean response matrix (time samples by number of directions) of each cell[37]. This decomposes the response into a temporal component and a direction-dependent component or tuning curve. An advantage of this procedure is that it does not require manual selection of time bins for computing direction tuning but extracts the direction-tuning curve given the varying temporal dynamics of different neurons.

To measure DS, we computed the vector sum in the 2D plane and used the resulting vector length as DS index. We additionally assessed the statistical significance of direction tuning using a permutation test[79]. Here, we created surrogate trials by shuffling the trial labels, computing the tuning curve and vector sum for each surrogate trial. Carrying out this procedure 1000 times generated a null distribution, assuming no direction tuning. We used the percentile of the true vector length as the $p$-value for the direction tuning.

*Clustering of GCL cells*: The preprocessed ROI traces of GCL cells ($n = 4519/8429$) were assigned to previously identified functional RGC clusters[37] by identifying for each cell the cluster with the best matching response properties. To account for a slight mismatch in frame rate for our stimulation systems compared to the previous one, calcium traces were shifted in time ($t = 40$ ms) and smoothed (for chirp stimulus only, boxcar smoothing with $n = 5$ points corresponding to 0.64 s) before calculating the linear correlation coefficients between a GCL cell's mean trace and all matching cluster mean traces for the chirp and the moving bar stimuli. Specifically, DS cells were correlated with DS clusters, non-DS cells were

correlated with non-DS clusters, and alpha cells (soma area > 170 μm²) were correlated with alpha cell clusters. To combine stimulus-specific correlations and response quality, we generated an overall match index (Mi) of each GCL cell to all RGC clusters[80]:

$$\text{Mi} = \frac{Qi_{Chirp}}{Qi_{Chirp} + Qi_{Bar}} * r_{Chirp} + \frac{Qi_{Bar}}{Qi_{Chirp} + Qi_{Bar}} * r_{Bar} \qquad (11)$$

Finally, each GCL cell with Mi > 0.5 was assigned to the cluster with the highest Mi and clusters were combined into functional RGC groups (for details, see ref. [37]).

**Statistical analysis**. We used the nonparametric Wilcoxon signed-rank test for quantifying the difference between cone surround responses under control and NBQX conditions (Fig. 2f), and field entropy of IPL and GCL scan fields (Supplementary Fig. 4).

We used the Chi-squared test to compare the distribution of anatomical and functional cone arrays (Supplementary Fig. 1h).

We used a Linear Mixed-Effects Model to analyze the difference between center and surround SC for OPL (Fig. 2b, c), IPL (Fig. 4b) and GCL recordings (Fig. 6b). This allowed to incorporate a random effects term in a linear predictor expression accounting for the fact that not all ROIs with a center response displayed a surround response (partially paired data). We used the lme4-package for R to implement the model and perform statistical testing[81]. For details, see Supplementary Methods.

We used Generalized Additive Models (GAMs) to analyze the relationship of difference in center and surround SC ($SC_{Diff}$) and IPL depth (Fig. 4e); opponency and IPL depth (Fig. 4f); center chromatic preference ($SC_{center}$) and IPL depth (Supplementary Fig. 2b). GAMs extend the generalized linear model by allowing the linear predictors to depend on arbitrary smooth functions of the underlying variables[82]. In practice, we used the mgcv-package for R to implement GAMs and perform statistical testing. For details, see Supplementary Methods.

To test if the number of color-opponent cells within single RGC and dAC groups was significantly larger/lower than expected from center SC and $SC_{Diff}$ we used a permutation test (Fig. 7g). First, we binned $SC_{Diff}$ values across all groups (bin size = 0.25). For every cell assigned to one group, we then randomly picked a different cell within the same $SC_{Diff}$ bin and with a similar center SC (±0.1). Like this, we generated an across-group distribution of $SC_{Diff}$ values with similar mean and s.d., but with shuffled cell labels. Then, we estimated the percentage of color-opponent cells ($\rho_{onset} < -0.3$ or $\rho_{offset} < -0.3$) in this across-group distribution and repeated this procedure for 10,000 times per group, generating a null distribution. Finally, we used the percentile of the true percentage of color-opponent cells as the p-value. We performed this analysis separately for RGCs (groups 1–32) and dACs (groups 33–46).

**Reporting summary**. Further information on research design is available in the Nature Research Reporting Summary linked to this article.

## Data availability
All relevant data are available at https://doi.org/10.5281/zenodo.3760607.

## Code availability
Data analysis was performed using IGOR Pro and R. Custom-written scripts are available at https://github.com/frankelab/retina_color.

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

## Acknowledgements

We thank G. Eske for excellent technical support, S. Haverkamp and T. Baden for feedback and discussions and J. Marvin and L. Looger for providing the viral vector (pAAV.hSyn.iGluSnFR). This work was supported by the German Federal Ministry of Education and Research (BMBF: FKZ: 01GQ1002 and 01GQ1601), the German Research Foundation (DFG: BE5601/4; SFB1233, project number 276693517; EXC 2064, project number 390727645; SCU 2243/3-1), and the Max Planck Society (MPG: M.FE.A. KYBE0004).

## Author contributions

K.F. designed the study with input from T.E. D.D. produced the iGluSnFR virus; K.F. performed viral injections; M.K. performed OPL recordings with help from K.F. and T.S. K.F. performed IPL recordings; K.S. performed GCL recordings with help from K.F. Y.R. performed electrical recordings; M.K., K.F., and K.S. performed preprocessing; K.F. analyzed the data with input from T.E. and P.B.; K.F. prepared the figures with help from K.S. and M.K. M.K., K.F., K.S., P.B., and T.E. wrote the manuscript.

## Competing interests

The authors declare no competing interests.
