## [Peer Review File · Nature Communications]

Reviewers' Comments:

Reviewer #1:

Remarks to the Author:

In the current study Kzatko et al. have used two-photon glutamate and calcium imaging at three distinct anatomical layers of mouse retina to systematically determine the origin of color opponency and how such color-opponent signals are processed by downstream bipolar and ganglion cells. A recent behavioral study by Denman et al. had shown that mice can only discriminate colors in their upper visual field but a complete understanding of the underlying mechanism remained unknown. Kzatko et al. have used a powerful optical imaging approach to carefully measure evoked responses to spectrally and spatially separate blue and green light. They discovered that centre-surround color opponency was mostly pronounced in the neurons of the ventral retina and it was present already at the outer synaptic layer in the cone output. They further show that color opponency was more prevalent in the OFF than ON bipolar cells. Lastly, ganglion cells show more complex diversity of chromatically tuned responses with cell type-specificity. Overall the study is a comprehensive and rigorous analysis which provides novel insights into the circuit mechanisms underlying color vision in mouse. The paper is well-written, data is well presented and analyzed. I have the following concerns:

- i) The authors suggest a rod-cone opponent mechanism mediated by the horizontal cells for color opponency observed in the ventral cones. It would be helpful to explain how such a rod based opponent mechanism is prevalent only in the ventral cones and not in dorsal cones. Does the HC connection to rods and cones differ in dorsal and ventral retina? Could this be assayed in a Nrl mutant mice where there are no rods?
- ii) Use of an electric lens for axial scanning of the inner plexiform layer has been mentioned by the authors. Please elaborate. What was the axial resolution and how separable are the ROIs in the z-axis?
- iii) Given the predominance of color opponent cells in the ventral retina and type-specificity at the level of ganglion cells, it would be great to directly show the comparison of color opponency for a given type like the slow ON Ganglion cell type, ON alpha, between dorsal and ventral retina.
- iv) While pharmacology can be difficult to interpret and non-specific, but given the generality of these findings across cells in the ventral retina, application of inhibitory receptor blocker such as Strychnine, might provide some insight into contribution of amacrine cell mediated inhibition in creating the centre-surround color opponency both for bipolar and ganglion cell output.
- v) Please provide a y-axis of dF/F for the fluorescence traces in figure 1 and 2.
- vi) In Fig. 2E, NBQX application causes an increase in the baseline fluorescence. Please explain.

Reviewer #2:

Remarks to the Author:

The present study, by Szatko et al. entitled "Neural circuits in the mouse retina support color vision in the upper visual field", is the first systematic attempt to study color processing across all layers and cell types in the mouse retina. This well-crafted study presents evidence that in mice, the ventral retina has evolved to relay chromatic information to the brain via different channels. By expressing iGluSnFR or by electroporating retinas with OGB-1, the authors sample the response properties using a well-established multiphoton imaging approach. This allowed them to show that chromatic

processing already starts in the photoreceptors, likely by combining information of cones and rods via horizontal cells - consistent with experiments performed in other species. Szatko et al. continued their study by examining how this chromatic information is further processed via the intricate networks in the retina. They describe that (i) the chromatic cone responses match the expected retinal opsin expression and that (ii) the chromatic coding varies dependent on the depth imaged depth in the IPL. The chromatic responses in the receptive field centers of bipolar cells in the ventral retina seem exclusively UV sensitive. In contrast, the ON layers have a contribution of S- and Rod-opsin. Finally, by imaging the GCL, the authors were capable of identifying a series of different ganglion cell types that would be capable of relaying information to the brain. Interestingly, the expected responses for different ganglion cells do not match the naïvely predicted responses based on the bipolar cell chromatic inputs. Thus, the authors suggest non-linear mechanisms involved in generating the measured ganglion cell responses.

Overall, this is a useful and interesting study that broadens our functional view on the visual system of the currently most prominent model organisms for vision research. However, I have one major technical concern that might change the interpretation, if proven right. As described in a review by the authors and cited in this study (Euler et al. 2019, preprints), multiphoton excitation light can excite the rod photoreceptors up to 2×10^4 R*/s group. This value is a back-on-the-envelope calculation that assumes of exponential decay of the rod photoreceptor absorption templates at ~ 940 nm. Thus, no one would be surprised if the value would vary by one order of magnitude up or down. However, this would change the interpretation of the results, since the higher isomerization value would saturate rods (at least until enough opsin is bleached). Next, I will describe why I sense that this might be indeed the case. There is little color-opponency in the "center" stimulus in cone and BP cells in the ventral retina. I would have assumed that this center stimulus should also encompass input from HC that are located in that "center" region since HC axonal processes do not show any particular directionality. Thus, an alternative hypothesis could be that the multiphoton excitation beam might saturate rods. Since the authors only scan a small ROI in the visually stimulated "center" region and avoid scanning the "surround", one could expect that the multiphoton light saturates only the rods localized in the "center". If this is the case, no rod signals will be measured in that region. I have seen the Suppl. Fig. 9; however, it would be better to show the same experiment with flashes. Spike triggered average requires long recordings that might change the adaptation of the retina compared to the flashes performed in most of the photoreceptor and bipolar cell experiments. In line with the adaptation concern, the spike-triggered averages in Suppl. Fig. 9 were likely taken with higher refresh rates than the bipolar event-triggered experiments (@ 10Hz) and for a shorter time. In the method section, the authors state that "green-sensitive cones were affected by the background illumination" (line 641-642). A statement about rods should be include.

To be comparable, the control experiments that are shown in Suppl. Fig. 9 should be repeated with flashes. I would also focus the laser on the different layers using the laser intensities used for the iGluSnFR. Another experiment that could uncover a possible saturation effect would be the following. Scan the entire visually stimulated area, using the same laser intensity per area, and test to what extent the surround responses are suppressed to flashes in cones/bipolar cells (or any variation of that idea).

Also, rod saturation could be an alternative explanation of the differences between the expected RGC color opponent cell numbers for ON and OFF RGC and why no rod signals could be measured. Since both experiments are done using different indicators, one synthetic (highly fluorescence, less laser power?) and one genetically encoded (probably dimmer, thus more laser power?). It would be helpful to know what laser intensities were used during all experiments. This wasn't mentioned in the methods. Was the laser intensity tuned according to the penetration depth? Did it change across experiments?

Understanding the extent by which rods are indeed differentially affected by the multiphoton light important. It has been shown in the past that mouse rods and cones interact via gap junctions (e.g., Astreti et al. 2014, Elife) and that rods can signal directly to at least some OFF BP cells (e.g., Soucy et al., 1998, Neuron). Assuming any of these pathways are active, classical center-surround color processing would require a bipolar cell pathway that avoids rod. Cone to BP cell selectivity is discussed; the rod to BP cell selectivity should also be discussed (e.g., Soucy et al., 1998, Neuron & Tsukamoto et al. 2001, J. Neurosci.).

Finally, I would have a suggestion regarding the discussion. I would include a short paragraph/sentence adding the recent results found invertebrates that show that color processing also occurs first in the photoreceptors (e.g., Schnaitmann et al. 2018, Cell). This could add a nice extra evolutionary spin to one of the main findings.

There is a typo in the citation of Barlow (line 562).

Reviewer #3:

Remarks to the Author:

This is a nice study that is well-written and clear using state-of-the-art imaging techniques to analyze responses from thousands of retinal neurons. They present clear and convincing evidence for the presence of many color opponent responses in the ventral, but not dorsal, retina of mouse. Prior to this, there had been very little published evidence for color opponency in mouse, largely consisting of a small population of melanopsin-containing M5 cells and cells located in the transitional zone of central mouse retina where there is strong co-expression of both S and M opsins. The finding that color opponency is largely limited to the ventral retina is thus surprising. The data also provide a comprehensive description of retinal cell types, from cones to ganglion cells, participating in color vision in this model animal species that will be valuable for future investigators trying to decode the pathways involved in this process. I offer a few questions and suggestions that might solidify a couple of their conclusions and clarify things for other readers.

1) Did the authors find S/M opponent responses in the transitional zone that had been described previously by Breuninger et al., 2011?

2) If the surround is mediated by rods, one would predict that it should disappear at lower frequencies than the center response since rods cannot follow fast flickering stimuli as effectively as cones. With the chirp stimulus (or sinusoidal stimuli), did the authors observe such a thing? That would provide nice direct evidence for their hypothesis.

3) The authors suggest that the absence of surround opponency in UV-sensitive cones might be due to light scatter into the center. I suggest they measure the amount of light scatter directly. One way they could do this is to place an autofluorescent plastic slide (available from Chroma) on the stage and stimulating it with the different center and surround lights. With these data, they could calculate whether the relative intensity of the scattered light is sufficient to activate the center. As well as providing a direct test of their hypothesis, this would be useful information for themselves and others regarding their center-surround resolution.

Minor comments:

1) What is the x/y and z-axis resolution of the microscope? What is the pixel size? How do these relate to the voxel size?

2) 4 of the 5 center/opponent ganglion cells shown in Suppl. Fig. S5 appear to be S-On cells. Do these

correspond to the S-ON/L-M OFF cells found in a number of other species? One would expect to encounter only a few true S-ON cells since true S cones are quite sparse.

3) In Fig. 7 A, the definitions of the bars are not clear to me. Are there only a few opponent ganglion cells and many non-opponent cells? I think that is the case, but is it instead that there is a slightly larger number of opponent cells than non-opponent cells? Please clarify in the legend.

4) There is a typo in the reference on line 562 (Barlow and BH, 1961). It should simply be (Barlow, 1961).

Signed, Wallace Thoreson

Reviewer #1 (Remarks to the Author):

In the current study Kzatko et al. have used two-photon glutamate and calcium imaging at three distinct anatomical layers of mouse retina to systematically determine the origin of color opponency and how such color-opponent signals are processed by downstream bipolar and ganglion cells. A recent behavioral study by Denman et al. had shown that mice can only discriminate colors in their upper visual field but a complete understanding of the underlying mechanism remained unknown. Kzatko et al. have used a powerful optical imaging approach to carefully measure evoked responses to spectrally and spatially separate blue and green light. They discovered that centre-surround color opponency was mostly pronounced in the neurons of the ventral retina and it was present already at the outer synaptic layer in the cone output. They further show that color opponency was more prevalent in the OFF than ON bipolar cells. Lastly, ganglion cells show more complex diversity of chromatically tuned responses with cell type-specificity. Overall the study is a comprehensive and rigorous analysis which provides novel insights into the circuit mechanisms underlying color vision in mouse. The paper is well-written, data is well presented and analyzed.

1.1 We thank the reviewer for appreciating our study and for the constructive comments.

I have the following concerns:

i) The authors suggest a rod-cone opponent mechanism mediated by the horizontal cells for color - opponency observed in the ventral cones. It would be helpful to explain how such a rod based opponent mechanism is prevalent only in the ventral cones and not in dorsal cones. Does the HC connection to rods and cones differ in dorsal and ventral retina? Could this be assayed in a *Nrl* mutant mice where there are no rods?

1.2 As correctly pointed out by the reviewer, our data suggest that the green surround component of ventral cones, which predominantly express UV-sensitive S-opsin (Röhlich et al., Neuron 1994), is mediated by horizontal cells driven by rod photoreceptors (cf. Fig. 2; see also Joesch and Meister, Nature 2016). The strong difference in spectral preference between the ventral cones' S-opsin-mediated center and rhodopsin-mediated surround results in color-opponent responses in these neurons.

Horizontal cells form highly stereotypical connections with cones and rods and, to our knowledge, there is no evidence that these connections systematically vary across the retina. Hence, it is likely that rods also contribute in a similar way to the surround of cones located in the dorsal retina. However, since most dorsal cones express green-sensitive M-opsin (Röhlich et al., Neuron 1994) with a peak sensitivity close to that of rhodopsin (cf. Fig. 1a), a surround mediated by rods would not result in color-opponency of these neurons.

*Isolating the contribution of rods and M-cones to the surround of dorsal cones would indeed be interesting, but is challenging because of the very similar spectral sensitivity profiles of M-opsin and rhodopsin in wild-type mice. *Nrl*-mutant mice appear to be an interesting option, however, there, the morphology of horizontal cell dendrites and axon terminals is hypertrophic and underdeveloped, respectively (Raven et al., J. Neurosci. 2007), which is expected to have an effect on cone chromatic processing. A better option would be *Opn1mw^R* mice, which express human L-opsin instead of mouse M-opsin (Smallwood et al., PNAS 2003), allowing to spectrally separate rods and M-cones with all retinal circuitry intact (e.g. Allen et al., Current Biology 2017). However, we do not have access to these mice.*

We now discuss the role of rods in generating surround responses of dorsal cones in the discussion of the revised manuscript.

ii) Use of an electric lens for axial scanning of the inner plexiform layer has been mentioned by the authors. Please elaborate. What was the axial resolution and how separable are the ROIs in the z-axis?

1.3 We quantified the axial resolution of x-z scans in a previous study by measuring the point spread function (PSF) of fluorescent beads (cf. Fig. 2D in Zhao et al., bioRxiv 2019). We found that the PSF was approx. 0.4 and 1.8 μm in the x-y plane and along the z-axis, respectively. ROIs in x-z scans were determined using a correlation-based approach (for details, see Methods and Zhao et al., bioRxiv 2019). This approach was very similar to that we recently used for x-y scans of the IPL (Franke et al., Nature 2017), where we demonstrated that single ROIs estimated by image correlation capture the signal of individual BC axon terminals.

To test whether the lower resolution along the z- compared to the y-axis had an effect on ROI estimation, we compared the distribution of ROI sizes in x-y and x-z scans (Rebuttal Fig. R1). We found that ROI areas were only slightly larger in x-z scans, but overall still covered the range expected for BC axon terminals (2-8 μm^2 ; cf. Extended Data Fig. 1 in Franke et al., Nature 2017). This indicates that the slightly lower separability of pixels along the z-axis in axial x-z scans compared to horizontal x-y scans has only a small effect on ROI separation.

In the revised manuscript, we have added a few sentences to the Methods section stating the axial resolution of the microscope and discussing ROI separability in x-z scans.

Rebuttal Fig. R1 | Comparison of ROI sizes in x-y and x-z scans of the IPL. Distribution of ROI areas in x-y (black; data from Franke et al., Nature 2017) and x-z scans (red; data from Zhao et al., bioRxiv 2019).

iii) Given the predominance of color opponent cells in the ventral retina and type-specificity at the level of ganglion cells, it would be great to directly show the comparison of color opponency for a given type like the slow ON Ganglion cell type, ON alpha, between dorsal and ventral retina.

1.4 We thank the reviewer for this suggestion. We added the distribution of center and surround chromatic preference and full-field opponency across the retina for two example RGC groups to Figure 7: On alpha RGCs (G_{24}) and On slow RGCs (G_{27}). We agree that this nicely illustrates the retinal location-specific differences in chromatic processing for individual RGC types.

iv) While pharmacology can be difficult to interpret and non-specific, but given the generality of these findings across cells in the ventral retina, application of inhibitory receptor blocker such as Strychnine, might provide some insight into contribution of amacrine cell mediated inhibition in creating the centre-surround color opponency both for bipolar and ganglion cell output.

1.5 We agree with the reviewer that it would be interesting to investigate the role of different AC circuits in generating BC and RGC color-opponency using pharmacology. In particular, it would be interesting to test whether the resulting drug effects are consistent across the population of BC and RGC types or specific for individual cell types.

We have started addressing these questions at the level of the BC output by recording glutamate release of ventral BCs across the IPL in response to UV and green center-

surround flicker, while pharmacologically blocking different subclasses of inhibitory neurons (Rebuttal Fig. R2).

First, we blocked spiking wide-field ACs using the voltage-gated sodium channel blocker TTX. We found that this eliminated the antagonistic response to UV surround stimulation for both On and Off BCs (Rebuttal Fig. R2c; surround strength ≤ 0). In contrast, the BCs' antagonistic response to green surround stimulation was only slightly affected by blocking wide-field ACs (surround strength ≤ 0). This finding is in line with our previous data, suggesting that the green surround component of BC receptive fields is generated by horizontal cells in the outer retina. To confirm that this was the case, we next used the GABA receptor blockers gabazine and TPMPA, which block both GABAergic wide-field ACs and the GABA-mediated component of horizontal cell feedback (Cueva et al., *J. Comp. Neurol.* 2002; Hirano et al., *eNeuro* 2016). This eliminated both the UV and the green component of BC surround, indicating that the green surround mediated by horizontal cells in the outer retina employs a GABA-dependent pathway. So far, we have not focused on glycinergic, small-field ACs as they mainly modulate the surround of BCs (Franke et al., *Nature* 2017).

This preliminary data demonstrates that blocking specific sub-populations of retinal neurons using pharmacology allows dissecting the circuits modulating chromatic processing of BCs and RGCs. However, we think that a systematic investigation of the type-specific roles of different interneurons in BC and RGC chromatic responses is beyond the scope of this study.

Rebuttal Figure 2 | Effect of TTX and Gbz/TPMPA on BC surround responses. **a**, Schematic illustrating the effect of Tetrodotoxin (TTX) on wide-field ACs. **b**, Stimulus kernels of an exemplary region-of-interest (ROI), estimated from a 10 Hz UV and green center and surround flicker stimulus. Dashed line indicates time point of response. **c**, Effect of TTX on UV and green surround strength (absolute kernel area multiplied by correlation with UV center kernel; $n=371/2/1$ ROIs/scan fields/mice). **d-f**, Like (a-c), but for Gbz/TPMPA (gabazine/(1,2,5,6-tetrahydropyridin-4-yl) methylphosphinic acid). Data from $n=110/2/1$ ROIs/scan fields/mice. Note that in addition to blocking GABAergic wide-field ACs, this also blocks GABA-mediated feedback from HCs to cones.

v) Please provide a y-axis of dF/F for the fluorescence traces in figure 1 and 2.

1.6 We scale the fluorescent traces according to standard deviation of the baseline. We have now added a scale bar for the y-axis in these figures.

vi) In Fig. 2E, NBQX application causes an increase in the baseline fluorescence. Please explain.

1.7 To evaluate the role of horizontal cell feedback in cone chromatic processing, we pharmacologically blocked light-mediated feedback from horizontal cells using the AMPA/KA-type glutamate receptor antagonist NBQX. Application of NBQX has been shown to eliminate tonic glutamatergic input to horizontal cells and, thereby, hyperpolarizes them (cf. Fig. 2 in Chapot et al., Current Biology 2017). As a consequence, inhibitory feedback to cones is reduced. This disinhibition increases the baseline Ca^{2+} level in cones, which has been demonstrated in mouse retinal slices (e.g. Kemmler et al., J. Neurosci. 2014), and results in an increase of baseline glutamate release.

We now added this information to the respective Results section.

Reviewer #2 (Remarks to the Author):

The present study, by Szatko et al. entitled "Neural circuits in the mouse retina support color vision in the upper visual field", is the first systematic attempt to study color processing across all layers and cell types in the mouse retina. This well-crafted study presents evidence that in mice, the ventral retina has evolved to relay chromatic information to the brain via different channels. By expressing iGluSnFR or by electroporating retinas with OGB-1, the authors sample the response properties using a well-established multiphoton imaging approach. This allowed them to show that chromatic processing already starts in the photoreceptors, likely by combining information of cones and rods via horizontal cells - consistent with experiments performed in other species. Szatko et al. continued their study by examining how this chromatic information is further processed via the intricate networks in the retina. They describe that (i) the chromatic cone responses match the expected retinal opsin expression and that (ii) the chromatic coding varies dependent on the depth imaged depth in the IPL. The chromatic responses in the receptive field centers of bipolar cells in the ventral retina seem exclusively UV sensitive. In contrast, the ON layers have a contribution of S- and Rod-opsin. Finally, by imaging the GCL, the authors were capable of identifying a series of different ganglion cell types that would be capable of relaying information to the brain. Interestingly, the expected responses for different ganglion cells do not match the naïvely predicted responses based on the bipolar cell chromatic inputs. Thus, the authors suggest non-linear mechanisms involved in generating the measured ganglion cell responses.

2.1 We thank the reviewer for appreciating our study and for the detailed and constructive comments.

Overall, this is a useful and interesting study that broadens our functional view on the visual system of the currently most prominent model organisms for vision research. However, I have one major technical concern that might change the interpretation, if proven right. As described in a review by the authors and cited in this study (Euler et al. 2019, preprints), multiphoton excitation light can excite the rod photoreceptors up to 2×10^4 R*/s group. This value is a back-on-the-envelope calculation that assumes of exponential decay of the rod photoreceptor absorption templates at ~ 940 nm. Thus, no one would be surprised if the value would vary by one order of magnitude up or down. However, this would change the interpretation of the results, since the higher isomerization value would saturate rods (at least until enough opsin is bleached).

2.2 We agree with the reviewer that the value for the rod photoreceptor activation by the 2-photon (2P) excitation laser represents only an estimate that is based on several assumptions, such as the number of rhodopsin molecules per rod and the assumed 2-photon cross-section of rhodopsin. However, even if 2P laser-induced rod activation was somewhat stronger than estimated in our previous studies, we do not think that this would substantially change the interpretation of our results.

The main reason for this is that in recent years, the classical view that (mouse) rod photoreceptors saturate at photopic light intensities ($>10^5$ R/s) has been challenged repeatedly (as discussed in Kelber, Curr Biol 2018). In the most systematic study so far, Tikidji-Hamburyan and colleagues (Tikidji-Hamburyan et al., Nat Comms 2017) have demonstrated that mouse rods progressively escape saturation for photopic light levels (up to 10^7 R*/s) over a period of 10-30 minutes, thereby driving robust visual responses in RGCs and neurons of the lateral geniculate nucleus (LGN) across a wide range of scotopic and photopic light intensities. Similarly, rod photoreceptors in guinea pig recover from saturation when presented with photopic background light levels (Demontis et al., Prog Brain Res 1993; Yin et al., J Neurosci 2006), suggesting that this might be a general feature of mammalian rods. Two recent studies investigated the potential mechanisms underlying the recovery of mouse rod responses. They found that a blue light-dependent pigment regeneration process in the photoreceptor membrane increases rod photosensitivity in bright light (Kaylor et al.,*

Nat Comms 2017) and that sensitization of the photovoltage by rod inner segment conductances extends the operating range of rods after bleaching (Pahlberg et al., J Physiol 2017).

That notion that mouse rods function at photopic light levels is further supported by the fact that (i) robust rod-driven ERG responses can be obtained at bright light conditions ($>10^5$ R/s; Pasquale et al., J Neurosci 2020), (ii) rods drive circadian photoentrainment across different photopic light levels (Altimus et al., Nat Neurosci 2010), (iii) cone-deficient mice can behaviorally discriminate contrasts at background light levels $>10^5$ R*/s (Naarendorp et al., J Neurosci 2010) and (iv) rod bipolar cells show robust light-correlated activity in the photopic regime (Chen et al., J Neurophysiol 2014; Franke et al., Nature 2017).*

Together, this strongly suggests that mouse rod photoreceptors are not saturated in the low photopic regime used in our imaging experiments. In the revised manuscript, we now reference the above mentioned studies. To further investigate the effect of the 2-photon laser on rod responses, we performed additional experiments (see below).

Next, I will describe why I sense that this might be indeed the case. There is little color-opponency in the "center" stimulus in cone and BP cells in the ventral retina. I would have assumed that this center stimulus should also encompass input from HC that are located in that "center" region since HC axonal processes do not show any particular directionality. Thus, an alternative hypothesis could be that the multiphoton excitation beam might saturate rods. Since the authors only scan a small ROI in the visually stimulated "center" region and avoid scanning the "surround", one could expect that the multiphoton light saturates only the rods localized in the "center". If this is the case, no rod signals will be measured in that region.

2.3 As argued above, recent evidence strongly suggests that mouse rods are not saturated at the laser intensities used in our experiments. To confirm that experimentally, we tested the effect of the 2P laser on rod-mediated surround responses of BCs. For that, we recorded BC glutamate release in response to UV and green center and surround flashes in a small horizontal IPL plane ($\sim 30 \times 30 \mu\text{m}$) in the ventral retina while increasing the size of the laser-illuminated area, until covering nearly the complete surround stimulus (Rebuttal Fig. R3). For our standard IPL recordings, the scan field only partially covers the center stimulus (100 μm in diameter). If the 2P laser was indeed saturating the rods, we would expect to see a strong decrease in green surround responses upon extending the laser-illuminated area into the cells' surround. However, we found that changing the laser-illuminated area had little effect on BC activity, including green surround responses. Again, this suggests that rods are not saturated by the 2P laser in our experiments.

Rebuttal Figure R3 | Effect of laser-illuminated area on chromatic bipolar cell responses. a, Schematic illustrating the size of the 2-photon laser-illuminated area (red) relative to center and surround stimuli and mean glutamate traces ($n=10$ trials in grey) of a small area ($30 \times 30 \mu\text{m}$) within the On sublamina of the inner plexiform layer (IPL) in response to center and surround UV and green flashes. **b,** Mean glutamate traces from (a) superimposed. **c,** Like (b), but for three additional scan fields.

If rods covered by the scan field are indeed not saturated, one would expect to see at least some degree of color-opponency in cone center responses - as pointed out by the reviewer. In our previous analysis, we have not systematically considered the frequency of center opponent responses at the level of the cone output. Therefore, we determined the number of ventral cones that showed color-opponent center responses. We found 49 of 1,337 ventral cones (originating from $n=7$ scan fields, $n=4$ mice) displaying reliable center opponent responses (Rebuttal Fig. R4a). However, for all these cones, antagonistic center responses were much weaker than respective antagonistic surround responses. We think the main reason why center opponency is relatively rare and weak at the level of the cone output is that most ventral cones did not only respond with a decrease in glutamate release to a UV center spot but showed a smaller decrease in glutamate upon green center stimulation (Rebuttal Fig. R4b). This is in line with the fact that the majority of cones in the ventral retina co-express both S- and M-opsin (e.g. Applebury et al., Neuron 2000). Therefore, an antagonistic center response (increase in glutamate) mediated by horizontal cells driven by rods might be masked by the hyperpolarizing response mediated by M-opsin activation. As center opponency is rare at the level of the cone output (see above) and most BC types sample from 3-8 cones within their dendritic tree (Behrens et al., eLife 2016), center opponency was not obvious at the level of the BC output.

Together, this suggests that rods are not saturated by the 2-photon laser and that center stimulation can be sufficient to drive rod-mediated color opponent responses in cones. In the revised manuscript, we now include the data on the effect of the 2P laser on rod-mediated surround responses (new Suppl. Fig. S10) and mention center opponency in cones.

Rebuttal Figure R4 | Center opponent responses of cones. a, Mean glutamate release of two exemplary cones located in the ventral retina exhibiting antagonistic, depolarizing responses (increase in glutamate release) to a green center and surround spot, likely due to HC input driven by rods. Dotted line indicates baseline. **b,** Mean glutamate traces of two exemplary ventral cones displaying a hyperpolarizing response (decrease in glutamate) to a green center spot, likely due to direct activation of M-opsin.

I have seen the Suppl. Fig. 9; however, it would be better to show the same experiment with flashes. Spike triggered average requires long recordings that might change the adaptation of the retina compared to the flashes performed in most of the photoreceptor and bipolar cell experiments. In line with the adaptation concern, the spike-triggered averages in Suppl. Fig. 9 were likely taken with higher refresh rates than the bipolar event-triggered experiments (@ 10Hz) and for a shorter time. In the method section, the authors state that "green-sensitive cones were affected by the background illumination" (line 641-642). A statement about rods should be include.

To be comparable, the control experiments that are shown in Suppl. Fig. 9 should be repeated with flashes. I would also focus the laser on the different layers using the laser intensities used for the iGluSnFR. Another experiment that could uncover a possible saturation effect would be the following. Scan the entire visually stimulated area, using the same laser intensity per area, and test to what extent the surround responses are suppressed to flashes in cones/bipolar cells (or any variation of that idea).

2.4 We thank the reviewer for these suggestions. To characterize chromatic responses of both BCs and RGCs, we have used a UV and green center-surround flicker stimulus (cf. Figs. 3 and 5). Consequently, to test the effect of the 2P laser on chromatic RGC responses (cf. Suppl Fig. S9), we displayed the exact same RGC flicker stimulus (5 Hz flicker for 10 minutes) during electrical recordings. Flashes were only used for cone recordings, because the slow frame rate of OPL scans (3.9 Hz) did not allow resolving responses to fast flickering stimuli. We agree with the reviewer that the long recording times (6 and 10 minutes for BC and RGC recordings, respectively) necessary for the flicker stimuli might have resulted in a different degree of adaptation compared to that for flash stimuli of shorter duration.

Therefore, we have performed new electrical recordings using UV and green center and surround flash stimuli (<1.5 minutes duration), similar to the stimulus used for cone recordings (cf. Fig. 1). In addition, we have varied the focal plane of the 2P laser while electrically recording from single RGCs - as suggested by the reviewer. For all new experiments, we set the laser power to the level we normally use for iGluSnFR experiments (~8 mW; see also 2.5 below). Other than for the flicker stimuli, where we saw a slight, laser-induced reduction in green center responses (cf. Suppl Fig. S9), we found no systematic effect of the 2P laser on the RGC flash responses, independent on the focal plane of the laser (Rebuttal Fig. R5). This suggests that for short durations of laser exposure, the 2P laser has no obvious effect on RGC chromatic responses. We have added this new data to

Suppl. Fig. S9. In addition, we added a new section to the Methods part (“Laser-induced effects on retinal activity”), where we discuss the effect of the 2P laser on retinal activity in more detail, with a focus on rod responses.

Rebuttal Figure R5 | Effect of imaging plane on chromatic retinal ganglion cell responses. a, Schematic illustrating the position of the laser focus across the retinal layers (left) and spiking activity and spike rate (in Hz) of an exemplary RGC (right) in response to UV and green center and surround flashes. Ctrl: without laser. **b, c,** Like (a), but for two additional cells. **d,** Quantification of change in firing rate ($n=5$ cells, $n=3$ animals) for the conditions shown in (a).

In summary, we think the following reasons argue against the view that rods might be saturated by the 2P laser:

- 1) Evidence from the last years strongly suggests that (mouse) rods are functional at the photopic light levels used in our experiments (for details, see 2.2). This has been demonstrated across different levels of the visual system, ranging from rod-driven BC (Chen et al., *J Neurophysiol* 2014; Franke et al., *Nature* 2017) and RGC responses (Tikidji-Hamburyan et al., *Nat Comms* 2017) in the isolated retina to rod-mediated visually guided behavior (Naarendorp et al., *J Neurosci* 2010; Pasquale et al., *J Neurosci* 2020), and using a variety of techniques, like electrophysiology, ERG and transgenic animals.
- 2) If rods are saturated by the laser, extending the laser-illuminated area into the surround should decrease rod-mediated color opponent surround responses of BCs. However,

new experiments demonstrated that BC surround responses were not affected by the laser (Rebuttal Fig. R3), suggesting that rods are functional and not saturated in our experiments.

- 3) In line with this, new analysis revealed that a low percentage of cones show color opponent center responses (Rebuttal Fig. R4), likely mediated by rod input to horizontal cells. As the center stimulus is approx. as large as OPL scan fields and, therefore, does barely activate any photoreceptors not illuminated by the 2P laser, this would not be observed if rods were saturated by the laser.*
- 4) Finally, new electrophysiological RGC recordings demonstrated that relatively short durations of laser exposure (<1.5 minutes) have no obvious effect on RGC chromatic responses (Rebuttal Fig. R5), independent on the imaging plane. Together with our previous experiments (cf. Suppl. Fig. S9), this suggests that for our experimental conditions adaptation of rods and cones due to the 2P laser contributed little to chromatic processing of retinal neurons.*

Also, rod saturation could be an alternative explanation of the differences between the expected RGC color opponent cell numbers for ON and OFF RGC and why no rod signals could be measured. Since both experiments are done using different indicators, one synthetic (highly fluorescence, less laser power?) and one genetically encoded (probably dimmer, thus more laser power?). It would be helpful to know what laser intensities were used during all experiments. This wasn't mentioned in the methods. Was the laser intensity tuned according to the penetration depth? Did it change across experiments?

2.5 In our experience, iGluSnFR (at the expression levels we routinely achieve) does not require a higher laser power than OGB-1. Hence, in our experiments, laser power largely depended on how deep in the retina the imaging plane was located and ranged between 7-13 mW. This meant that we used lower laser powers for GCL scans (~7 mW) compared to IPL (~9 mW) and OPL scans (~12 mW). The increased laser powers for IPL and OPL scans was required to compensate for power loss due to scattering of the 2P laser within the tissue. Additional adjustments of the laser power to compensate for differences in dye labeling or indicator expression were typically within ± 1 mW.

While we cannot exclude that these differences in laser power led to changes in retinal adaptation, we consider it as highly unlikely that this substantially affected our results -- also in view of the new control measurements in Rebuttal Figs. R3, R5.

In the revised manuscript, we added the information about the laser powers used in the different experiments to the Methods section.

Understanding the extent by which rods are indeed differentially affected by the multiphoton light is important. It has been shown in the past that mouse rods and cones interact via gap junctions (e.g., Astreti et al. 2014, Elife) and that rods can signal directly to at least some OFF BP cells (e.g., Soucy et al., 1998, Neuron). Assuming any of these pathways are active, classical center-surround color processing would require a bipolar cell pathway that avoids rod. Cone to BP cell selectivity is discussed; the rod to BP cell selectivity should also be discussed (e.g., Soucy et al., 1998, Neuron & Tsukamoto et al. 2001, J. Neurosci.).

2.6 Here we disagree with the reviewer -- rod-cone mediated center-surround color-opponency does not require BCs receiving excitatory inputs exclusively from cones. Red-green color opponency in the primate retina is a nice example demonstrating that selective wiring to spectrally distinct photoreceptor types is not a prerequisite for generating color opponent center-surround RFs (Crook et al., J Neurosci 2011; Field et al., Nature 2010; Buzas et al., J Neurosci 2006; Martin et al., Nature 2001). These studies have shown that a different ratio

of M-/L-cone input to center and surround of midget RGCs is sufficient to generate color-opponency. Similarly, retinal neurons in the S-opsin dominated ventral retina will display color-opponent responses if the ratio of rod/cone input differs for their center and surround. Therefore, while direct rod input to BCs or signal spread from rods to cones via gap junctions will reduce the difference in rod/cone input ratio of center and surround, it likely will still yield center-surround color-opponency. Note that at the low photopic light intensities used in our experiments, rod-cone coupling is thought to be very weak (Ribelayga et al., Neuron 2008; Jin et al., J Physiol 2015).

We now mention these points more prominently in the Discussion of the revised manuscript.

Finally, I would have a suggestion regarding the discussion. I would include a short paragraph/sentence adding the recent results found invertebrates that show that color processing also occurs first in the photoreceptors (e.g., Schnaitmann et al. 2018, Cell). This could add a nice extra evolutionary spin to one of the main findings.

2.7 Thanks for this suggestion. In the revised manuscript, we now added a few sentences to the Discussion about how our results relate to findings in invertebrates.

There is a typo in the citation of Barlow (line 562).

2.8 Changed.

Reviewer #3 (Remarks to the Author):

This is a nice study that is well-written and clear using state-of-the-art imaging techniques to analyze responses from thousands of retinal neurons. They present clear and convincing evidence for the presence of many color opponent responses in the ventral, but not dorsal, retina of mouse. Prior to this, there had been very little published evidence for color opponency in mouse, largely consisting of a small population of melanopsin-containing M5 cells and cells located in the transitional zone of central mouse retina where there is strong co-expression of both S and M opsins. The finding that color opponency is largely limited to the ventral retina is thus surprising. The data also provide a comprehensive description of retinal cell types, from cones to ganglion cells, participating in color vision in this model animal species that will be valuable for future investigators trying to decode the pathways involved in this process. I offer a few questions and suggestions that might solidify a couple of their conclusions and clarify things for other readers.

3.1 We thank the reviewer for appreciating our study and helping to improve the manuscript.

1) Did the authors find S/M opponent responses in the transitional zone that had been described previously by Breuninger et al., 2011?

3.2 We assume the reviewer is referring to the study by Chang et al. (Neuron 2013), where the authors investigated the effect of the asymmetric opsin distribution in mice on chromatic processing of RGCs. They find that RGCs located in the opsin transitional zone show color-opponent responses due to chromatically distinct input to their center and surround.

As mentioned in the Discussion, our findings are in line with this study, showing that many RGCs located in the opsin transitional zone are color-opponent (Fig. 6b, c). In addition, our data demonstrate that color-opponency is not restricted to the opsin transitional zone, but an abundant feature of RGCs located in the ventral retina, likely generated by a rod-cone opponent circuit (see also Joesch and Meister, Nature 2016). Together, this suggests that there are different mechanisms underlying color-opponency in the mouse retina and that their relative contribution changes with retinal position.

In the revised manuscript, we now mention this point more prominently in the Discussion.

2) If the surround is mediated by rods, one would predict that it should disappear at lower frequencies than the center response since rods cannot follow fast flickering stimuli as effectively as cones. With the chirp stimulus (or sinusoidal stimuli), did the authors observe such a thing? That would provide nice direct evidence for their hypothesis.

3.3 Thanks for this suggestion. To directly test if BC surround responses likely mediated by rods are tuned to lower frequencies than cone-mediated center responses, we recorded BC glutamate release in response to sinusoidal modulation (2, 4, 6, 8, and 10 Hz) of a UV center spot and green surround annulus (Rebuttal Fig. R6a). For each ROI, we then estimated a center and surround frequency tuning curve by computing the fundamental response component for each modulation frequency (Rebuttal Fig. R6b). We found that BC responses to a UV center spot were indeed tuned to somewhat higher temporal frequencies compared to responses elicited by a green surround annulus (Rebuttal Fig. R6b). Under the assumption that rods cannot follow high frequencies as efficiently as cones (e.g. Umino et al., J Neurosci 2008, Wang et al., J Neurosci 2011), this would indeed support our hypothesis that green surround responses in ventral BCs are mediated by rods.

It is noteworthy that the difference in centroid frequency between BC center and surround responses was relatively small (center: 4.69 ± 1.18 Hz; surround: 3.98 ± 0.84 Hz). Partially, this may be due to the fact that estimating the fundamental response component is sensitive to noise, which may have resulted in underestimating the difference in temporal tuning of BC center and surround. However, the relative small difference in centroid frequency

resonates with a new study published only a few weeks ago: It suggests that mouse rods may signal much faster changes than previously thought, probably by providing direct inputs to Off cone BCs and/or via glycinergic ACs (Pasquale et al., J Neurosci 2020).

Rebuttal Figure R6 | Frequency tuning of bipolar cell center and surround responses. *a*, Mean glutamate responses of exemplary region-of-interest (ROI) in response to a UV center (left, black) and green surround (right, grey) sine stimulus modulated at different frequencies. Stimulus trace on the top shows 2 Hz sine modulation. Traces were recorded at a frame rate of 62.5 Hz. *b*, Normalized tuning curve of traces shown in (a), with centroid (center of mass) indicated for each condition. *c*, Centroids of all ROIs ($n=10$ ROIs, $n=5$ scan fields, $n=2$ mice) for center and surround condition. $p < 0.05$, Wilcoxon-signed rank test.

3) The authors suggest that the absence of surround opponency in UV-sensitive cones might be due to light scatter into the center. I suggest they measure the amount of light scatter directly. One way they could do this is to place an autofluorescent plastic slide (available from Chroma) on the stage and stimulating it with the different center and surround lights. With these data, they could calculate whether the relative intensity of the scattered light is sufficient to activate the center. As well as providing a direct test of their hypothesis, this would be useful information for themselves and others regarding their center-surround resolution.

3.4 Thanks for this suggestion. To quantify the spatial resolution of our projector, we have recently established an alternative approach to the autofluorescent slides mentioned by the reviewer. Specifically, we have visualized checkerboards of different checker sizes using a raspberry pi camera chip positioned at the level of the recording chamber (cf. Fig. 4 in Franke et al., eLife 2019). Our measurements confirmed that the spatial resolution of our projector is similar for UV and green stimuli, resolving checkers of sizes $>5 \mu\text{m}$. Therefore, we think that the resolution of the stimulator does not contribute to the observed effect in cones.

Instead, we hypothesize that scattering of UV light within the retina, combined with the higher sensitivity of S- compared to M-opsin (Baden et al., Neuron 2013), may have contributed to our finding that UV-sensitive cones did not show an antagonistic response to a UV surround spot. However, we did not come up with a suitable experimental approach for assessing this experimentally. Therefore, we decided to phrase our hypothesis more carefully in the revised manuscript.

Minor comments:

1) What is the x/y and z-axis resolution of the microscope? What is the pixel size? How do these relate to the voxel size?

3.5 We quantified the axial and horizontal resolution of our microscope in a previous paper by measuring the point spread function (PSF) of fluorescent beads (cf. Fig. 2D in Zhao et al., bioRxiv 2019; see also our reply 1.3). We found that the PSF was approx. 0.4 in the x-y plane and $1.8 \mu\text{m}$ along the z-axis. We have added this information to the Methods section.

2) 4 of the 5 center/opponent ganglion cells shown in Suppl. Fig. S5 appear to be S-On cells. Do these correspond to the S-ON/L-M OFF cells found in a number of other species? One would expect to encounter only a few true S-ON cells since true S cones are quite sparse.

3.6 This is a misunderstanding. S-On/M-Off cells, which have been described in a number of mammalian species, were very rare in our dataset (1 of 5 cells shown in Suppl. Fig. S5). Most of the center-opponent RGCs in our dataset were S-Off/M-On cells (4 of 5 cells shown in Suppl. Fig. S5). Such a response profile could be generated by an amacrine cell that inverts the signal of S-On type 9 BCs, as demonstrated in ground squirrel (Sher and DeVries, Nat Neurosci 2012; Chen and Li, Nat Neurosci 2012) and rabbit (Mills et al., J Neurosci 2014). It would be very interesting to investigate whether a similar mechanism exists in the mouse retina, e.g. by pharmacologically blocking specific amacrine cell populations (see our reply 1.5).

3) In Fig. 7 A, the definitions of the bars are not clear to me. Are there only a few opponent ganglion cells and many non-opponent cells? I think that is the case, but is it instead that there is a slightly larger number of opponent cells than non-opponent cells? Please clarify in the legend.

3.7 Thanks for pointing this out. We now added the description of the bars to the legend of Figure 7.

4) There is a typo in the reference on line 562 (Barlow and BH, 1961). It should simply be (Barlow, 1961).

3.8 Changed.

Signed, Wallace Thoreson

Reviewers' Comments:

Reviewer #1:

Remarks to the Author:

The authors have addressed most of my concerns.

I have two comments on the author's response to the concern about saturation of rods (raised by reviewer2) with the two-photon excitation.

- In a recent study, Grimes et al., eLife 2018 (Fig 3), using suction electrode recordings show that 90% of the circulating current in mouse rods is suppressed at a light level of 1000R*/rod/sec. Perhaps the authors could speculate if they think it is the inner segment conductances as suggested by Pahlberg et al., J Physiol 2017, that allows the rods to weakly respond at the photopic light levels.
- Have the authors made their recordings in isolated retina or retina attached to the underlying pigment epithelium (which would allow pigment regeneration)? If the former then the authors should be careful while comparing rod saturation in their preparation with the observations made by Naarendorp et al., J Neurosci 2010 and Pasquale et al., J Neurosci 2020 on rod-mediated behavior in intact animals.

Reviewer #2:

Remarks to the Author:

The revisions provided by the authors were thoroughly done and convincing. I don't have anything else to add and would recommend this work for publication.

Reviewer #3:

Remarks to the Author:

This well-written paper attempts a comprehensive assessment of color processing through the different layers of the mouse retina. They largely succeed in this effort, limited only by some unavoidable technical issues (e.g., alignment of the stimulus to the center of the recording field rather than the center of each cell's receptive field). Their results suggest a fairly robust representation of color in the ventral retina, derived from an opponency between rod and cone responses. Similar to red/green opponency in primates, this opponency arises at the first synapse and does not require cone-type specific wiring mechanisms. The data are strong and the authors have addressed my few concerns. I believe this paper will have an important impact on the field.

Reviewers' Comments

Reviewer #1 (Remarks to the Author):

The authors have addressed most of my concerns.

I have two comments on the author's response to the concern about saturation of rods (raised by reviewer 2) with the two-photon excitation.

- In a recent study, Grimes et al., eLife 2018 (Fig 3), using suction electrode recordings show that 90% of the circulating current in mouse rods is suppressed at a light level of 1000R*/rod/sec. Perhaps the authors could speculate if they think it is the inner segment conductances as suggested by Pahlberg et al., J Physiol 2017, that allows the rods to weakly respond at the photopic light levels.

Thanks for this comment. As mentioned in our reply 2.2 in the previous rebuttal, two recent studies investigated the potential mechanisms underlying the recovery of mouse rod responses. They found that a blue light-dependent pigment regeneration process in the photoreceptor membrane increases rod photosensitivity in bright light (Kaylor et al., Nat Comms 2017) and that sensitization of the photovoltage by rod inner segment conductances extends the operating range of rods after bleaching (Pahlberg et al., J Physiol 2017). This suggests that both inner and outer segment might contribute to the recovery of rod responses. We now mention these citations in the Discussion. Whether similar mechanisms exist in primate rod photoreceptors remains to be determined. In Grimes et al., eLife 2018, rod responses were recorded over a relatively short time period, which might not be sufficient to recover from saturation. It will be interesting to see whether time-dependent recovery of saturation also exists in other mammalian species – as shown for guinea pig (Demontis et al., Prog Brain Res 1993; Yin et al., J Neurosci 2006).

- Have the authors made their recordings in isolated retina or retina attached to the underlying pigment epithelium (which would allow pigment regeneration)? If the former then the authors should be careful while comparing rod saturation in their preparation with the observations made by Naarendorp et al., J Neurosci 2010 and Pasquale et al., J Neurosci 2020 on rod-mediated behavior in intact animals.

Our recordings were performed in isolated retinas with pigment epithelium detached. In mice, the pigment epithelium easily detaches from the retina and therefore, experiments with pigment epithelium attached are challenging. We agree with the reviewer that the degree of photoreceptor saturation critically depends on whether an intact pigment epithelium is present. We did not want to directly compare our results to behavioral studies mentioned by the reviewer. Instead, we mention these studies in the Discussion of our manuscript to provide evidence that mouse rod photoreceptors are active at low photopic conditions. We now added that experiments with isolated retinas lack intact pigment epithelium to improve clarity.

Reviewer #2 (Remarks to the Author):

The revisions provided by the authors were thoroughly done and convincing. I don't have anything else to add and would recommend this work for publication.

Reviewer #3 (Remarks to the Author):

This well-written paper attempts a comprehensive assessment of color processing through the different layers of the mouse retina. They largely succeed in this effort, limited only by some unavoidable technical issues (e.g., alignment of the stimulus to the center of the recording field rather than the center of each cell's receptive field). Their results suggest a fairly robust representation of color in the ventral retina, derived from an opponency between rod and cone responses. Similar to red/green opponency in primates, this opponency arises at the first synapse and does not require cone-type specific wiring mechanisms. The data are strong and the authors have addressed my few concerns. I believe this paper will have an important impact on the field.